# Heterogeneous Data Game: Characterizing the Model Competition Across Multiple Data Sources

**Renzhe Xu** [1 2]   **Kang Wang** [2]   **Bo Li** [3]

## Abstract

Data heterogeneity across multiple sources is common in real-world machine learning (ML) settings. Although many methods focus on enabling a single model to handle diverse data, real-world markets often comprise multiple competing ML providers. In this paper, we propose a game-theoretic framework—the *Heterogeneous Data Game*—to analyze how such providers compete across heterogeneous data sources. We investigate the resulting pure Nash equilibria (PNE), showing that they can be non-existent, homogeneous (all providers converge on the same model), or heterogeneous (providers specialize in distinct data sources). Our analysis spans monopolistic, duopolistic, and more general markets, illustrating how factors such as the "temperature" of data-source choice models and the dominance of certain data sources shape equilibrium outcomes. We offer theoretical insights into both homogeneous and heterogeneous PNEs, guiding regulatory policies and practical strategies for competitive ML marketplaces.

## 1. Introduction

Data heterogeneity is commonplace in real-world machine learning (ML) applications, where data often originate from multiple sources with distinct distributions (Li et al., 2017; Hendrycks et al., 2020; Gulrajani & Lopez-Paz, 2021; Liu et al., 2023). For example, in health care, patient data may be gathered from different hospitals, each serving varied

demographics and disease prevalences. Such heterogeneous settings arise across diverse fields, including the digital economy and scientific research.

Much of the existing literature on heterogeneous data focuses on devising a single ML method that performs robustly across all data sources (Arjovsky et al., 2019; Kuang et al., 2020; Liu et al., 2021b; Duchi & Namkoong, 2021). However, real-world markets typically have multiple ML providers (Black et al., 2022; Jagadeesan et al., 2023a), each aiming to optimize its performance relative to others. For instance, competing diagnostic tool providers offer models to hospitals, which then choose a provider based on local performance criteria. This competitive interplay differs significantly from single-provider frameworks (Nisan et al., 2007) and can lead to market dynamics unaddressed by previous approaches.

Several works (Ben-Porat & Tennenholtz, 2017; 2019; Feng et al., 2022; Jagadeesan et al., 2023a; Iyer & Ke, 2024; Einav & Rosenfeld, 2025) have analyzed competition among multiple ML model providers, examining Nash equilibria, social welfare, and agents' strategies under competition. However, these studies mainly focus on a single data distribution and do not account for heterogeneity across multiple sources.

In this paper, we develop a game-theoretic framework to study multiple providers competing over heterogeneous data sources. We then analyze the resulting pure Nash equilibria to uncover how data heterogeneity and competitive forces shape providers' strategies.

### 1.1. Overview of the Heterogeneous Data Game

We introduce the *Heterogeneous Data Game* to model the competition among multiple ML model providers across diverse data sources. Consider $K$ distinct data sources, each associated with a weight $w_k$ representing its proportion, and joint distributions $P_k(x, y)$ over features $x$ and labels $y$. In this market, each of the $N$ ML model providers selects a model parameterized by $\hat{\theta}_n$. Following previous works on model and platform competition (Jagadeesan et al., 2023a; Drezner & Eiselt, 2024), the utility of each model provider is determined by its market share across the different data sources. Specifically, each data source $k$ selects an ML

---
[1]MoE Key Laboratory of Interdisciplinary Research of Computation and Economics, Shanghai University of Finance and Economics, China. [2]Institute for Theoretical Computer Science, Shanghai University of Finance and Economics, China. [3]School of Economics and Management, Tsinghua University, China. Emails: xurenzhe@sufe.edu.cn, wangkang0330@gmail.com, libo@sem.tsinghua.edu.cn. Correspondence to: Bo Li <libo@sem.tsinghua.edu.cn>.

*Proceedings of the 42nd International Conference on Machine Learning*, Vancouver, Canada. PMLR 267, 2025. Copyright 2025 by the author(s).

|  | Heterogeneous Data Game under the *Proximity* Choice Model | Heterogeneous Data Game under the *Probability* Choice Model |
|---|---|---|
| Monopoly ($N = 1$) | The single model chooses the parameter given by Eq. (9). | |
| Duopoly ($N = 2$) | Equivalent condition for PNE existence (Thm. 5.1) 
 PNE must be heterogeneous, if it exists (Thm. 5.1) | Equivalent condition for PNE existence (Thm. 5.2) 
 PNE must be homogeneous, if it exists (Thm. 5.2) |
| $N > 2$ | Sufficient condition for PNE existence (Thm. 5.4 and Cor. 5.5) 
 PNE must be heterogeneous, if it exists (Prop. 5.3) | Equivalent condition for homogeneous PNE existence (Thm. 5.6) 
 Sufficient condition for heterogeneous PNE existence (Thm. 5.7) 
 Example when both types of PNE exist simultaneously (Ex. 5.2) |

*Table 1.* Overview of the results.

model based on the observed losses of available models. A provider's utility is then the sum of $w_k$ from all data sources that adopt its model. Consequently, each provider strategically chooses $\hat{\theta}_n$ to maximize its utility.

Motivated by linear models, we represent each data source with two statistics: a ground-truth parameter $\theta_k$ for $P_k(y|x)$ and a covariance matrix $\Sigma_k$ for $P_k(x)$. From a distribution-shift perspective, variations in $\theta_k$ and $\Sigma_k$ across sources correspond to concept shift and covariate shift, respectively—two common types of distribution shifts in practice (Liu et al., 2021b). Additionally, the loss of a model $\hat{\theta}_n$ on data source $k$ is calculated as the squared Mahalanobis distance, $(\hat{\theta}_n - \theta_k)^\top \Sigma_k (\hat{\theta}_n - \theta_k)$, corresponding to the mean squared error (MSE) in linear model settings.

For data sources' choice models, we adopt two standard frameworks (Drezner & Eiselt, 2024): the *proximity choice model* (Hotelling, 1929; Plastria, 2001; Ahn et al., 2004), where each data source selects the provider with the lowest loss (with ties broken uniformly), and the *probability choice model* (Wilson, 1975; Hodgson, 1981; Bell et al., 1998), where data sources may choose sub-optimal models based on a logit framework (Train, 2009), controlled by a temperature parameter $t$.

### 1.2. Overview of the Results

An overview of these results is presented in Tab. 1. We investigate the pure Nash equilibria (PNE) of the Heterogeneous Data Game and identify three patterns of PNEs across different game setups: (1) *Non-existence of PNE.* In this case, no PNE exists, leading to an unstable ML model market. (2) *Homogeneous PNE.* Here, all model providers independently train their ML models to minimize the $w_k$-weighted loss across all data sources. As a result, this type of PNE leads to the homogeneity of models available in the market. (3) *Heterogeneous PNE.* In this scenario, model providers offer different ML models. Most specialize in a single data source, typically adopting the ground-truth parameter $\theta_k$ of a specific data source $k$.

**Monopoly** ($N = 1$). In this setting, a single provider can achieve the same utility with any ML model parameter. However, it typically chooses the parameter that minimizes

the weighted loss across all data sources, denoted by $\hat{\theta}^{\mathrm{M}}$.

**Duopoly** ($N = 2$). Under the proximity choice model, we specify conditions for the existence of a PNE and show that, if a PNE exists, both providers choose the ground-truth parameter of the data source with the maximal weight. In contrast, under the probability choice model, any PNE must be homogeneous, with both providers choosing $\hat{\theta}^{\mathrm{M}}$, the parameter that minimizes the weighted loss across sources.

**More than two providers** ($N > 2$). Under the proximity choice model, if a PNE exists, providers tend to pick different models, leading to a heterogeneous PNE. Moreover, when a few data sources have significantly larger weights (Kairouz et al., 2021; Li et al., 2020), a PNE exists if $N$ lies within a certain range, and providers fully specialize in those dominant sources. In contrast, under the probability choice model, both homogeneous and heterogeneous PNE may arise, depending on the temperature $t$. Specifically, when $t$ is small, indicating that data sources are highly unlikely to choose sub-optimal models, only a heterogeneous PNE may exist. Conversely, when $t$ is large, meaning data sources are more likely to uniformly choose among all available models, only a homogeneous PNE may exist. We also present an example where both types of PNE exist simultaneously.

Our theoretical findings yield several insights for multi-provider ML markets. First, they illuminate how the interplay of data heterogeneity, choice models, and competition can produce either homogeneous or heterogeneous equilibria, thereby influencing the variety of models offered. Second, they indicate that when a few data sources dominate, providers tend to specialize in those sources, potentially overlooking smaller ones; this outcome calls for appropriate incentive mechanisms. Finally, market parameters—such as the temperature in the probability choice model—can be adjusted by market regulators to foster either heterogeneous model offerings or convergence toward homogeneous solutions. Taken together, these insights can inform both regulatory policy and practical strategies for building competitive ML marketplaces.

## 2. Related Works

**Data heterogeneity.** In real-world scenarios, data often exhibit significant heterogeneity due to variations in time, space, and population during the data collection process (Liu et al., 2023). The concept of data heterogeneity has been extensively studied across multiple disciplines, including ecology (Li & Reynolds, 1995), economics (Rosenbaum, 2005), and computer science (Wang et al., 2019). This work focuses on the implications of data heterogeneity in machine learning settings. In this context, considerable research has aimed to ensure that a single model performs robustly across diverse test environments (Liu et al., 2021b), leading to a range of effective methodological frameworks, including causal learning (Bühlmann, 2020; Peters et al., 2016), invariant learning (Arjovsky et al., 2019; Liu et al., 2021a; Koyama & Yamaguchi, 2020), stable learning (Xu et al., 2022; Kuang et al., 2020; Yu et al., 2023), and distributionally robust optimization (Sinha et al., 2018; Duchi & Namkoong, 2021; Liu et al., 2022). However, these existing approaches largely overlook the presence of multiple competing model providers and the strategic interactions that arise in such settings.

**Competition in machine learning.** Our work extends prior research on competition among machine learning model providers under homogeneous data settings (Ben-Porat & Tennenholtz, 2017; 2019; Feng et al., 2022; Jagadeesan et al., 2023a; Einav & Rosenfeld, 2025). Specifically, Ben-Porat & Tennenholtz (2017; 2019) studied best-response dynamics and algorithmic methods for finding pure Nash equilibria (PNE) in regression tasks, while Einav & Rosenfeld (2025) extended these insights to classification. Feng et al. (2022) explored the bias–variance trade-off in competitive environments, showing that competing agents tend to favor variance-induced error over bias. Jagadeesan et al. (2023a) demonstrated that increasing model size does not necessarily improve social welfare. In contrast to these studies, we consider *heterogeneous* data sources with distinct distributions, uncovering novel equilibrium structures and establishing new conditions for their existence.

**Competitive location models.** Our framework is technically related to competitive location models (Hotelling, 1929; Shaked, 1975; d'Aspremont et al., 1979; Eiselt et al., 1993; Plastria, 2001; Ahn et al., 2004), as comprehensively surveyed by Drezner & Eiselt (2024). However, most existing models focus on low-dimensional spaces or networks with uniform distance metrics, largely due to two factors: (1) applications in urban planning naturally align with one-dimensional (Hotelling, 1929; d'Aspremont et al., 1979), two-dimensional (Tsai & Lai, 2005; Shaked, 1975; Lederer & Hurter Jr, 1986), or network-based (Eiselt & Laporte, 1991; 1993; Dorta-González et al., 2005) formula-

tions; and (2) many models incorporate additional variables such as price or quantity, which reduce tractability and restrict attention to small-scale settings. While a few studies investigate high-dimensional competition, they primarily address quantity competition (Anderson & Neven, 1990) or pricing (Bester, 1989), rather than spatial or parameter-based competition. By contrast, our setting considers source-specific distance metrics arising from distributional shifts, along with high-dimensional strategy spaces driven by a large number of data sources and model parameters. These distinctions introduce substantial challenges for theoretical analysis.

**Other competitive frameworks.** Finally, our work connects to competition scenarios in targeted advertising (Iyer & Ke, 2024; Iyer et al., 2024), online marketplaces (Liu et al., 2020; Hron et al., 2023; Jagadeesan et al., 2023b; Yao et al., 2024a;b), platform competition (Jullien & Sand-Zantman, 2021; Calvano & Polo, 2021), and broader game-theoretic analyses (Immorlica et al., 2011). Unlike these studies, we highlight how heterogeneous data distributions shape market equilibria among multiple ML model providers.

## 3. Heterogeneous Data Game (HD-Game)

### 3.1. Notations

We begin by introducing several essential notations. For a positive integer $N$, let $[N]$ denote the set $\{1, 2, \ldots, N\}$. The $N$-dimensional simplex, denoted by $\Delta_N$, is defined as $\Delta_N = \{(x_1, x_2, \ldots, x_N) : \sum_{i=1}^{N} x_i = 1 \text{ and } x_i \geq 0, \forall i \in [N]\}$. For any square matrix $A$, we use $A \succ 0$ to indicate that $A$ is positive definite, and $A \succeq 0$ to indicate that $A$ is positive semi-definite. Furthermore, given a positive definite square matrix $\Sigma \succ 0$, the Mahalanobis distance between two vectors $x$ and $y$ is defined as $d_M(x, y; \Sigma) = \sqrt{(x-y)^\top \Sigma^{-1}(x-y)}$.

### 3.2. Game Setup

**Heterogeneous data.** Consider a setting with $K \geq 2$ data sources. Each source $k$ has a true model parameter $\theta_k \in \mathbb{R}^D$ and a covariance matrix $\Sigma_k$. These two terms capture concept shift (via $\theta_k$) and covariate shift (via $\Sigma_k$), respectively (Liu et al., 2021b), as detailed in Sec. 3.3. We further assume $\theta_k \neq \theta_{k'}$ for all $k \neq k'$, since any two sources with identical parameters can be merged into one.

For a model parameterized by $\theta \in \mathbb{R}^D$, the loss associated with data source $k$ is defined as the squared Mahalanobis distance between $\theta$ and $\theta_k$ with $\Sigma_k^{-1}$, i.e., $d_M^2(\theta, \theta_k; \Sigma_k^{-1})$. As shown in Sec. 3.3, the Mahalanobis distance could correspond to the mean square error (MSE) of $\theta$ on data source $k$ in linear model settings, and it can measure the error caused by both concept shift and covariate shift.

Additionally, each data source $k$ is assigned a weight $w_k$, representing its proportion within the total data. Without loss of generality, we assume the weights are ordered and $w_1 > w_2 > \cdots > w_K > 0$, with $\sum_{k=1}^{K} w_k = 1$. Let $\boldsymbol{w} = (w_1, w_2, \ldots, w_K)$ denote the vector of weights.

**Model providers.** There are $N$ model providers (players)[1] that need to compete the models in these $K$ data sources. Each player $n \in [N]$ needs to choose one model $\hat{\theta}_n \in \mathbb{R}^D$, and the loss of player $n$ for data source $k$, denoted as $\ell_{n,k}$, is

$$\ell_{n,k} = d_M^2(\hat{\theta}_n, \theta_k; \Sigma_k^{-1}) = (\hat{\theta}_n - \theta_k)^\top \Sigma_k (\hat{\theta}_n - \theta_k). \quad (1)$$

**Data sources' choice model.** The data sources will choose which model to deploy based on the losses $\ell_{n,k}$. Formally, let $g : \mathbb{R}^N \to \Delta_N$ be the choice model. For a data source $k$, given $N$ losses $\ell_{1,k}, \ell_{2,k}, \ldots, \ell_{N,k}$, the function $g(\ell_{1,k}, \ldots, \ell_{N,k})$ will output an $N$-dimensional vector, and its $n$-th element, denoted as $g_n(\ell_{1,k}, \ldots, \ell_{N,k})$, is the probability of choosing the $n$-th model. Following previous works (Jagadeesan et al., 2023a; Drezner & Eiselt, 2024), we consider two types of choice models for estimating the market share of different participants:

- **Proximity choice model**. Here, each data source chooses the model with the least loss. When several models exhibit the same loss, the data source will randomly choose one model with equal probabilities. Formally,

$$g_n^{\text{PROX}}(\ell_{1,k}, \ldots, \ell_{N,k})$$
$$= \begin{cases} 0, & \text{if } \exists j \in [N], \ell_{j,k} < \ell_{n,k} \\ \frac{1}{|\{j \in [N]: \ell_{j,k} = \ell_{n,k}\}|}, & \text{otherwise.} \end{cases} \quad (2)$$

- **Probability choice model**. Following (Jagadeesan et al., 2023a), we assume that data sources may noisily choose the models based on the following logit model (Train, 2009),

$$g_n^{\text{PROP}}(\ell_{1,k}, \ldots, \ell_{N,k}) = \frac{\exp(-\ell_{n,k}/t)}{\sum_{j=1}^N \exp(-\ell_{j,k}/t)}. \quad (3)$$

with a temperature parameter $t > 0$. Intuitively, the parameter $t$ controls the willingness for each data source to choose sub-optimal models. When $t \to 0$, this model will become the proximity model as shown in Eq. (2). By contrast, when $t \to \infty$, all models become indifferent and the data source tends to choose models randomly.

[1]We use the terms "model provider" and "player" interchangeably.

**The Heterogeneous Data Game.** Given a strategy profile $\hat{\boldsymbol{\theta}} = (\hat{\theta}_1, \hat{\theta}_2, \ldots, \hat{\theta}_N)$, the utility of player $n$ is

$$u_n(\hat{\boldsymbol{\theta}}) = \sum_{k=1}^K w_k g_n(\ell_{1,k}, \ldots, \ell_{N,k}). \quad (4)$$

We note that for each $n \in [N]$, the term $\ell_{n,k}$ is implicitly a function of $\hat{\theta}_n$, as defined in Eq. (1). For any $\theta \in \mathbb{R}^D$, we use $(\theta, \hat{\boldsymbol{\theta}}_{-n})$ to denote the strategy profile in which player $n$ deviates from their original strategy $\hat{\theta}_n$ to a new strategy $\theta \in \mathbb{R}^D$. We focus on the properties of the Pure Nash Equilibrium (PNE), formally defined as follows.

**Definition 3.1** (Pure Nash Equilibrium (PNE)). A strategy profile $\hat{\boldsymbol{\theta}} = (\hat{\theta}_1, \ldots, \hat{\theta}_N)$ is a pure Nash equilibrium if, for all $n \in [N]$ and $\theta \in \mathbb{R}^D$, $u_n(\theta, \hat{\boldsymbol{\theta}}_{-n}) \leq u_n(\hat{\boldsymbol{\theta}})$.

In practice, model providers incur significant costs when retraining multiple models and typically deploy a single model rather than adopting a mixed strategy. Consequently, it is more realistic to analyze the pure Nash equilibrium (PNE), where each provider commits to a specific model, rather than the mixed Nash equilibrium (MNE), which assumes randomized selection among multiple models.

Note that the utility function depends on whether the choice model $g_n(\cdot)$ in Eq. (4) is set as $g_n^{\text{PROX}}$ or $g_n^{\text{PROP}}$, as defined in Eqs. (2) and (3). Consequently, different choice models yield different games and, therefore, different PNEs. For clarity, we refer to the heterogeneous data game under the proximity choice model as **HD-Game-Proximity** and under the probability choice model as **HD-Game-Probability** in the remainder of this paper.

### 3.3. Motivating Example — Linear Model

Consider a linear model setting with $K$ data sources, each associated with a distribution $P_k(x, y)$ for $k \in [K]$, where $x \in \mathbb{R}^D$ denotes the feature vector and $y \in \mathbb{R}$ is the corresponding label. Assume that $x$ is normalized such that $\mathbb{E}_{P_k}[x] = 0$. The covariance matrix under $P_k$ is then given by $\Sigma_k = \mathbb{E}_{P_k}[xx^\top] \succ 0$. Furthermore, assume that the conditional distribution $P_k(y \mid x)$ follows a linear model with parameter $\beta_k$, perturbed by Gaussian noise $\varepsilon \sim \mathcal{N}(0, \sigma_k^2)$; that is, $y \mid x \sim \mathcal{N}(\beta_k^\top x, \sigma_k^2)$.

Consider $N$ players, each selecting a parameter $\hat{\beta}_n$. The MSE of player $n$ on data source $k$ is given by

$$\mathbb{E}_{P_k}\left[\left(\hat{\beta}_n^\top x - y\right)^2\right] = \left(\hat{\beta}_n - \beta_k\right)^\top \Sigma_k \left(\hat{\beta}_n - \beta_k\right) + \sigma_k^2$$
$$= d_M^2\left(\hat{\beta}_n, \beta_k; \Sigma_k^{-1}\right) + \sigma_k^2. \quad (5)$$

It is easy to verify that the noise term in Eq. (5) does not affect the choice models in Eqs. (2) and (3). Consequently,

we confirm that using the squared Mahalanobis distance as a loss function aligns with the mean squared error (MSE), validating that the game effectively characterizes model provider competition in linear settings.

Moreover, linear probing—where only the final linear layer is updated—is a widely used technique for adapting pre-trained models to downstream tasks, particularly when fine-tuning the full model is computationally expensive or prone to overfitting (Kumar et al., 2022). Since features $x$ can represent either raw inputs or embeddings from pretrained models, our framework also extends to scenarios where model providers employ the linear probing technique.

## 4. Basic Properties and Assumptions

### 4.1. Basic Properties of HD-Game

We first characterize the possible strategy set in equilibria for each player.

**Proposition 4.1.** *Denote the set $\vartheta$ as follows:*

$$\vartheta \triangleq \left\{ \bar{\theta}(\boldsymbol{q}) : \boldsymbol{q} = (q_1, q_2, \ldots, q_K) \in \Delta_K \right\}, \quad (6)$$

*where*

$$
\begin{aligned}
\bar{\theta}(\boldsymbol{q}) &\triangleq \arg\min_{\theta} \sum_{k=1}^{K} q_k d_M^2 \left( \theta, \theta_k; \Sigma_k^{-1} \right) \\
&= \left( \sum_{k=1}^{K} q_k \Sigma_k \right)^{-1} \left( \sum_{k=1}^{K} q_k \Sigma_k \theta_k \right).
\end{aligned}
\quad (7)
$$

*Then, the following holds:*

- *In HD-Game-Proximity, if a PNE exists, there must be one where every player's strategy belongs to $\vartheta$.*

- *In HD-Game-Probability, any PNE necessarily requires all players' strategies to lie within $\vartheta$.*

*Remark* 4.1. Prop. 4.1 suggests that players will generally choose strategies from the set $\vartheta$, which corresponds to minimizing a weighted loss over data sources, with each player determining their respective weights.

### 4.2. Assumptions

We introduce the following regularity assumption.

**Assumption 4.1.** For any $\theta \in \mathbb{R}^D$, there is at most one $\boldsymbol{q} \in \Delta_K$ such that $\bar{\theta}(\boldsymbol{q}) = \theta$.

*Remark* 4.2. When $\Sigma_1 = \Sigma_2 = \cdots = \Sigma_K$, this assumption reduces to requiring that $\theta_1, \theta_2, \ldots, \theta_K$ be affinely independent. For general settings, the number of parameters $D$ is typically large, whereas the number of data sources $K$ is relatively small, often satisfying $K \leq D$. Consequently, this assumption is generally reasonable in real-world settings.

**Assumption 4.2.** For all $i, j, k \in [K]$ with $i \neq j$, $d_M(\theta_i, \theta_k; \Sigma_k^{-1}) \neq d_M(\theta_j, \theta_k; \Sigma_k^{-1})$.

*Remark* 4.3. This assumption ensures that no two ground-truth models $\theta_i$ and $\theta_j$ from distinct data sources have identical losses on a given data source $k$. Since different data sources typically have distinct ground-truth models, this condition is generally satisfied in practice.

**Problem-dependent constants.** We introduce the following constant based on the game's parameters:

$$\ell_{\max} \triangleq \max_{\theta \in \vartheta, k \in [K]} d_M^2(\theta, \theta_k; \Sigma_k^{-1}), \quad (8)$$

which represents the maximum possible loss for any strategy in $\vartheta$. Intuitively, $\ell_{\max}$ quantifies the degree of heterogeneity among data sources. A small $\ell_{\max}$ indicates that any model $\theta \in \vartheta$ incurs relatively low loss across all data sources, suggesting minimal variation among them. Conversely, a large $\ell_{\max}$ implies greater difficulty in finding a single model that performs well across all sources, representing higher data heterogeneity.

## 5. Pure Nash Equilibria Analysis

In this section, we formally characterize the pure Nash equilibria (PNE) of our Heterogeneous Data Game under the different data-source choice models introduced earlier. As previewed in the introduction, we establish three possible outcomes, each governed by distinct sufficient conditions:

1. **Non-existence of PNE.** In certain settings, no PNE arises, indicating that the model market remains fundamentally unstable. In other words, providers continually adjust their strategies in response to each other, preventing any long-term equilibrium.

2. **Homogeneous PNE.** Here, a stable equilibrium exists in which all model providers converge on the same parameter (e.g., the one minimizing the $w_k$-weighted loss across sources). This outcome yields a market dominated by essentially one "universal" model.

3. **Heterogeneous PNE.** Here, model providers differentiate themselves by specializing in distinct data sources. Typically, each provider adopts the ground-truth parameter $\theta_k$ of one source, resulting in a diverse array of models.

We observe that the concepts of "homogeneous PNE" and "heterogeneous PNE" are analogous to the ideas of "minimal" and "maximal" differentiation in location theory (Drezner & Eiselt, 2024). However, we maintain the terms "homogeneous" and "heterogeneous" because our distance metric differs across data sources.

Below, we present the theoretical results for each outcome in the contexts of monopoly (Sec. 5.1), duopoly (Sec. 5.2),

and general multi-provider markets (Sec. 5.3).

## 5.1. Monopoly Setting

When $N = 1$, the model choice is arbitrary, as there is no competition. However, the model provider typically selects a strategy that minimizes the overall loss across all data sources. Consequently, the chosen strategy, denoted as $\hat{\theta}^M$, is given by:

$$\hat{\theta}^M \triangleq \bar{\theta}(\boldsymbol{w}) = \arg\min_\theta \sum_{k=1}^K w_k d_M^2 \left(\theta, \theta_k; \Sigma_k^{-1}\right). \quad (9)$$

## 5.2. Duopoly Setting

In this subsection, we consider the duopoly setting where there are only 2 model providers in the market.

### 5.2.1. HD-GAME-PROXIMITY

**Theorem 5.1.** *Consider HD-Game-Proximity with $N = 2$, and suppose Assump. 4.1 holds. Then:*

1. *If $w_1 < 0.5$, a PNE does not exist.*

2. *If $w_1 \geq 0.5$, a PNE exists, and $\hat{\boldsymbol{\theta}} = (\theta_1, \theta_1)$ is a PNE. Moreover, if $w_1 > 0.5$, $(\theta_1, \theta_1)$ is the unique PNE.*

*Remark* 5.1. Thm. 5.1 shows that in HD-Game-Proximity with $N = 2$, a PNE exists if and only if $w_1 \geq 0.5$, indicating the presence of a dominant data source. Moreover, when $w_1 > 0.5$, both model providers specialize in the dominant data source by selecting $\theta_1$.

Although both providers adopt the same strategy, this PNE is still classified as heterogeneous, consistent with the general HD-Game-Proximity results in Sec. 5.3. Notably, unlike the homogeneous PNE in HD-Game-Probability, players in this PNE specialize in a single dominant data source rather than optimizing across all sources.

### 5.2.2. HD-GAME-PROBABILITY

**Theorem 5.2.** *Consider HD-Game-Probability with $N = 2$, and suppose Assump. 4.1 holds. If a PNE exists, then the only possible PNE is that both players choose $\hat{\theta}^M$ (defined in Eq. (9)).*

*Furthermore, there exists a constant $\underline{t} \leq 2\ell_{\max}$, depending on all game parameters, such that $\hat{\theta}^M$ is a PNE if and only if $t \geq \underline{t}$.*

*Remark* 5.2. Compared to Thm. 5.1, in HD-Game-Probability, a PNE may fail to exist even if $w_1 \geq 0.5$ when $t$ is smaller than the threshold $\underline{t}$. This may seem counterintuitive, as one might expect the probability choice model to converge to the proximity choice model as $t \to 0$. However, the properties of PNEs are not consistent in this limit. This inconsistency arises because, for $N = 2$, the only possible

PNE requires both players to choose $\hat{\theta}^M$, as established in the first part of this theorem. Notably, this inconsistency may not hold for $N > 2$, which we demonstrate in Thm. 5.7.

Deriving a closed-form expression for the threshold $\underline{t}$ is generally intractable. Empirically, we observe that $\underline{t} \approx C_0 \cdot (2\ell_{\max})$ with $0 < C_0 < 1$, where $C_0$ depends on the game's parameters. Experiments (see Sec. 6) consistently show that greater data-source heterogeneity—measured by $\ell_{\max}$ in Eq. (8)—pushes the threshold $\underline{t}$ upward. Hence, as heterogeneity grows, a homogeneous PNE can arise only when data sources exhibit an even stronger tendency to select sub-optimal models.

## 5.3. General Cases with More than Two Model Providers

In this subsection, we analyze ML model markets with more than two model providers.

### 5.3.1. HD-GAME-PROXIMITY

**Heterogeneity in PNE.** We first show that in HD-Game-Proximity, any existing PNE tends to be heterogeneous.

**Proposition 5.3.** *Consider HD-Game-Proximity, and suppose Assump. 4.1 holds. Let $\hat{\boldsymbol{\theta}} = \{\hat{\theta}_1, \ldots, \hat{\theta}_N\}$ be a PNE. For any $\theta \in \mathbb{R}^D$ such that $\theta \notin \{\theta_1, \ldots, \theta_K\}$, let $m = |\{j : \hat{\theta}_j = \theta\}|$. Then, $m \leq 1$.*

*Remark* 5.3. Prop. 5.3 shows that in HD-Game-Proximity, if a PNE exists, no two players will adopt the same model unless it corresponds to a ground-truth model of a data source. This suggests that model providers tend to offer distinct models, leading to a heterogeneous PNE.

Moreover, in some cases, achieving a PNE requires certain players to select strategies outside the set $\{\theta_1, \theta_2, \ldots, \theta_K\}$. A detailed example is provided in App. A.

**Sufficient conditions for the existence of heterogeneous PNE.** We derive sufficient conditions under which a heterogeneous PNE exists and can be explicitly characterized.

**Theorem 5.4.** *Consider HD-Game-Proximity and suppose that Assumps. 4.1 and 4.2 hold. Assume there exists a constant $k_0 \in [K]$ such that $w_{k_0} > 3\sum_{j=k_0+1}^K w_j$. Then PNE exists if $N$ satisfies*

$$\sum_{k=1}^{k_0} \left\lfloor \frac{3w_k'}{w_{k_0}'} \right\rfloor \leq N \leq \sum_{k=1}^{k_0} \left(\left\lceil \frac{w_k'}{\sum_{j=k_0+1}^K w_j} \right\rceil - 1\right) \quad (10)$$

*where for any $k$ such that $1 \leq k \leq k_0$,*

$$w_k' = w_k + \sum_{j=k_0+1}^K w_j \mathbb{1}\left[k = \arg\min_{1 \leq j' \leq k_0} d_M\left(\theta_{j'}, \theta_k; \Sigma_j^{-1}\right)\right].$$

$$(11)$$

*Moreover, for any PNE $\hat{\boldsymbol{\theta}} = (\hat{\theta}_1, \ldots, \hat{\theta}_N)$, it must hold that $\forall n \in [N], \hat{\theta}_n \in \{\theta_1, \theta_2, \ldots, \theta_{k_0}\}$. In addition, let $m_k = |\{j \in [N] : \hat{\theta}_j = \theta_k\}|$ be the number of players choosing strategy $\theta_k$ in the PNE. Then,*

$$\forall k \in [k_0], \left| m_k - \left\lfloor \frac{w'_k}{z^*} \right\rfloor \right| \leq 1 \qquad (12)$$

*where $z^* = \sup\left\{ z > 0 : h(z) \triangleq \sum_{k=1}^{k_0} \lfloor w'_k/z \rfloor \geq N \right\}$.*

*Remark* 5.4. Thm. 5.4 suggests that when a few data sources carry dominant weights—a phenomenon commonly observed in practice (Kairouz et al., 2021; Li et al., 2020)—and the number of model providers $N$ lies within a certain range, a PNE exists in which all providers specialize in serving a single data source. Moreover, in such equilibria, the number of providers allocated to each source is approximately proportional to its weight.

Concretely, the choice of $k_0$ indicates that the top-$k_0$ data sources hold significantly higher weights, each at least three times the total weight of the remaining data sources. The constraints on $N$ in Eq. (10) ensure that (1) providers consider the $k_0$-th data source and (2) data sources with small weights are overlooked. Under these conditions, the exact form of the PNE can be derived. In a PNE, all model providers select a ground-truth model from the top-$k_0$ data sources. Consequently, the utility of non-dominant data sources is assigned to the nearest dominant data source, as characterized by Eq. (11). Furthermore, since $z^*$ is fixed in Eq. (12), $m_k$ is proportional to $w'_k$. This aligns with intuition, as data sources with higher weights typically attract more model providers optimizing for them.

This insight implies that policymakers can mitigate imbalanced attention among different data sources by either introducing more providers or incentivizing focus on underrepresented sources. Thm. 5.4 provides a quantitative foundation for both interventions.

We further provide an example to explain Thm. 5.4.

**Example 5.1.** Consider a setting with $K = 4$ data sources and weights $\boldsymbol{w} = (0.6, 0.35, 0.03, 0.02)$. The ground-truth models are given by $\theta_1 = (1, 0, 0)$, $\theta_2 = (-1, 0, 0)$, $\theta_3 = (1, 0.1, 0)$, and $\theta_4 = (-1, 0, 0.1)$, with covariance matrices $\Sigma_1 = \Sigma_2 = \Sigma_3 = \Sigma_4 = I$. In this case, the first two data sources have dominant weights.

Setting $k_0 = 2$, a heterogeneous PNE is guaranteed to exist when $N$ lies within a specific range ($[8, 19]$ in this case). In this PNE, model providers will only select strategies from $\{\theta_1, \theta_2\}$. Since data source 3 has a similar ground-truth model to data source 1, and data source 4 to data source 2, providers selecting $\theta_1$ benefit from data source 3, while those selecting $\theta_2$ benefit from data source 4.

For instance, when $N = 10$, the PNE consists of six play-

ers choosing $\theta_1$ and four choosing $\theta_2$, approximately proportional to $(w'_1, w'_2)$, where $w'_1 = w_1 + w_3 = 0.63$ and $w'_2 = w_2 + w_4 = 0.37$.

In addition, Thm. 5.4 implies the following corollary.

**Corollary 5.5.** *Consider HD-Game-Proximity and suppose that Assumps. 4.1 and 4.2 hold. When $N \geq \sum_{k=1}^{K} \lfloor 3w_k/w_K \rfloor$, a PNE exists. Moreover, for any PNE $\hat{\boldsymbol{\theta}} = (\hat{\theta}_1, \ldots, \hat{\theta}_N)$, it holds that $\forall n \in [N], \hat{\theta}_n \in \{\theta_1, \theta_2, \ldots, \theta_K\}$. In addition, let $m_k = |\{j \in [N] : \hat{\theta}_j = \theta_k\}|$ be the number of players that choose strategy $\theta_k$ in the PNE. We have $\forall k \in [K], |m_k - \lfloor w_k/z^* \rfloor| \leq 1$ where $z^* = \sup\left\{ z > 0 : \sum_{k=1}^{K} \lfloor w_k/z \rfloor \geq N \right\}$.*

*Remark* 5.5. This result follows directly from Thm. 5.4 with $k_0$ set to $K$. It implies that a PNE always exists when $N$ is sufficiently large.

### 5.3.2. HD-GAME-PROBABILITY

Unlike HD-Game-Proximity, we show that both homogeneous and heterogeneous PNEs can exist in HD-Game-Probability.

**Homogeneous PNE.** We first derive equivalent conditions for the existence of a homogeneous PNE, as well as a sufficient condition for its uniqueness.

**Theorem 5.6.** *Consider HD-Game-Probability and suppose that Assump. 4.1 holds. Let $\hat{\boldsymbol{\theta}}^{Homo} = (\hat{\theta}^M, \hat{\theta}^M, \ldots, \hat{\theta}^M)$, where $\hat{\theta}^M$ is defined in Eq. (9). Then there exist two constants: $0 < \underline{t} \leq 2\ell_{\max}$, depending on all game parameters, and $C > 0$, depending only on $\{\Sigma_k, \theta_k, w_k\}_{k=1}^{K}$, such that the following results hold:*

1. *$\hat{\boldsymbol{\theta}}^{Homo}$ is a PNE if and only if $t \geq \underline{t}$.*
2. *If $t \geq \max\{6C/N, 2\ell_{\max}\}$, then $\hat{\boldsymbol{\theta}}^{Homo}$ is the unique PNE.*

*Remark* 5.6. Thm. 5.6 implies that a homogeneous PNE does not exist when $t$ is small. As $t$ increases, a homogeneous PNE may emerge, and for sufficiently large $t$, it becomes the unique PNE. The closed-form expressions of $\underline{t}$ and $C$ are difficult to derive. However, similar to Thm. 5.2, synthetic experiments in Sec. 6 suggest that the threshold $\underline{t}$ required for PNE existence is approximately $C_0 \times (2\ell_{\max})$, where $0 < C_0 < 1$ is a game-specific constant. Moreover, as $N$ increases, a homogeneous PNE is more likely to be unique, as indicated by the second part of Thm. 5.6. This trend is further verified by our synthetic experiments in Sec. 6.

**Heterogeneous PNE.** We next demonstrate that for sufficiently small $t$, a heterogeneous PNE can exist.

**Theorem 5.7.** *Consider HD-Game-Probability, and suppose that Assumps. 4.1 and 4.2 hold, with $N \geq$*

$\sum_{k=1}^{K} \lfloor 3w_k/w_K \rfloor$. *Additionally, assume that for all $n, n' \in [N]$ and distinct $k, k' \in [K]$, it holds that $w_k/n \neq w_{k'}/n'$. Let $\hat{\boldsymbol{\theta}}^{Prox} = (\hat{\theta}_1^{Prox}, \ldots, \hat{\theta}_N^{Prox})$ be a PNE in the corresponding HD-Game-Proximity. Then, there exists a constant $\underline{t} > 0$ such that for any $t \leq \underline{t}$, a PNE $\hat{\boldsymbol{\theta}}^{Hete} = (\hat{\theta}_1^{Hete}, \ldots, \hat{\theta}_N^{Hete})$ exists in HD-Game-Probability and satisfies*

$$\forall n \in [N], \quad \left\| \hat{\theta}_n^{Hete} - \hat{\theta}_n^{Prox} \right\|_2 \leq t^2.$$

*Remark* 5.7. For simplicity, we build on the conditions of Cor. 5.5. Thm. 5.7 establishes that when $t$ is sufficiently small, a heterogeneous PNE exists and closely approximates the PNE in the corresponding HD-Game-Proximity. This is expected, as HD-Game-Probability approaches HD-Game-Proximity for large $N$ as $t \to 0$. However, proving this result is significantly more challenging due to the continuous nature of the potential strategy set for each model provider.

Additionally, while the specified range for $N$ is sufficient, it is not necessary. Our experiments in Sec. 6 suggest that a heterogeneous PNE may also exist for smaller values of $N$.

Thms. 5.6 and 5.7 show that, to promote model diversity (i.e., the emergence of heterogeneous PNE) and prevent convergence to a single dominant model (i.e., homogeneous PNE), policymakers can increase the rationality of data sources—that is, enhance their ability to select higher-performing models—which, in turn, encourages the formation of heterogeneous PNE.

**Existence of both PNE at the same time.** We now present an example to illustrate that both types of PNE can coexist in a single game.

**Example 5.2.** Let $N = 8$ and $K = 2$ with $\theta_1 = (1, 1)$ and $\theta_2 = (0, 1)$. The covariance matrices are $\Sigma_1 = \Sigma_2 = I$, and the weights are $\boldsymbol{w} = (0.53, 0.47)$. The temperature parameter is set to $t = 0.4$. Since $K = 2$, Prop. 4.1 implies that each model provider selects a weight $\alpha_n \in [0, 1]$, yielding the final model $\hat{\theta}_n = \alpha_n \theta_1 + (1 - \alpha_n)\theta_2 = (\alpha_n, 1)$.

In this setting, we identify two types of PNE: (1) a homogeneous PNE, where all model providers choose $\alpha_n = 0.53$, and (2) a heterogeneous PNE, where four providers choose $\alpha_n \approx 0.76$ (type 1) and the other four choose $\alpha_n \approx 0.30$ (type 2). In the heterogeneous PNE, model providers specialize in different data sources, forming two distinct groups.

As shown in Fig. 1, we plot each player's utility if they deviate to a different policy. From the figure, it is evident that no player benefits from deviating, thereby verifying the correctness of the PNEs.

## 6. Synthetic Experiments

In this section, we conduct synthetic experiments to investigate how the temperature parameter $t$ in the probability

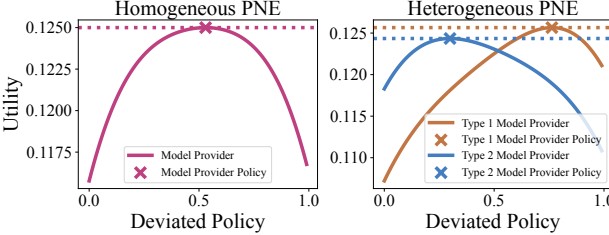

*Figure 1.* Utility of a single model provider with a deviated policy for both homogeneous and heterogeneous PNE in the configuration of Ex. 5.2.

choice model (Eq. (3)) influences the existence of homogeneous and heterogeneous PNEs in HD-Game-Probability.

**Data-generating processes.** Because our theoretical results do not depend on the number of data sources $K$, we set $K = 2$ for simplicity. We also choose $D = 2$ to fulfill Assump. 4.1. We randomly generate 10 game configurations with different $\{\Sigma_k, \beta_k, w_k\}_{k \in [K]}$. Each covariance matrix is constructed so that its largest eigenvalue does not exceed 1. In addition, we set $w_2 \geq 0.1$ to avoid a scenario where the first data source fully dominates the market. The number of model providers $N$ is varied from $\{2, 3, 4, \ldots, 30\}$ to explore the effect of $N$ on the critical values of $t$.

**Calculating critical temperature parameters.** Following Prop. 4.1, each model provider $n$'s strategy $\hat{\theta}_n$ must take the form $\bar{\theta}(\boldsymbol{q}_n)$ with $\boldsymbol{q}_n \in \Delta_2$ and $\boldsymbol{q}_n = (\alpha_n, 1 - \alpha_n)$ for $0 \leq \alpha_n \leq 1$. To verify whether a candidate strategy profile $\hat{\boldsymbol{\theta}}$ is a PNE, we enumerate all possible deviations: for each provider $n$, we check every $\alpha_n \in \{0, 0.002, 0.004, \ldots, 0.998, 1\}$ to see if a profitable deviation exists. Using this enumeration, we identify:

- **Homogeneous PNE.** We seek the threshold $\underline{t}$ given in Thms. 5.2 and 5.6. Specifically, we search over $\underline{t} \in \{0.001, 0.002, \ldots, 0.999, 1\} \times (2\ell_{\max})$ to find the minimal $t$ for which the strategy profile $\hat{\boldsymbol{\theta}}^{Homo}$ (from Thm. 5.6) is indeed a PNE.

- **Heterogeneous PNE.** We seek the maximal $t$ for which a heterogeneous PNE exists. Since determining the exact maximum can be complex in game theory (Gottlob et al., 2003; Fabrikant et al., 2004), we adopt an empirical procedure inspired by the proof of Thm. 5.7. For each candidate $t$, we use Alg. 2 (discussed in App. B) to obtain a heterogeneous PNE candidate, then apply the same enumeration technique to verify whether it is a PNE. We thus find the largest $t$ in $\{0.001, 0.002, \ldots, 0.999, 1\} \times (2\ell_{\max})$ for which our approach can produce a heterogeneous PNE. Although this does not guarantee the true maximum, it provides a useful lower bound.

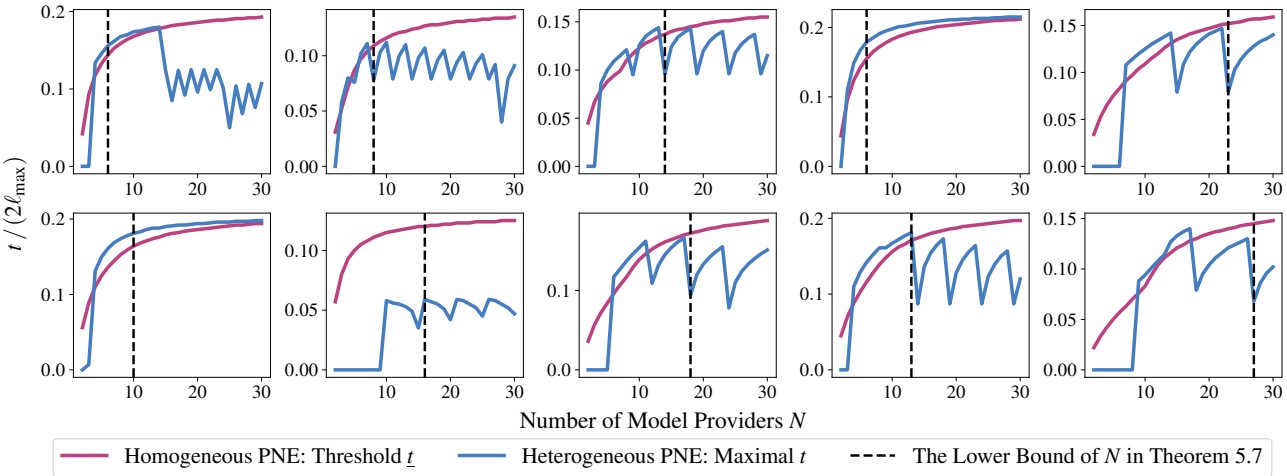

*Figure 2.* In the probability choice model, this figure reports, across 10 randomly generated games, the threshold $\underline{t}$ that guarantees the existence of a homogeneous PNE and the approximate largest value of $t$ that guarantees the existence of a heterogeneous PNE, as $N$ varies.

**Results and analysis.** Fig. 2 presents our experimental results. We make the following observations:

1. **Homogeneous PNE.** The threshold temperature $\underline{t}$ given in Thms. 5.2 and 5.6 generally increases with $N$, but the growth curve flattens for larger $N$. This is consistent with Thms. 5.2 and 5.6, which guarantee that $\underline{t} \leq 2\ell_{\max}$ and thus ensure the existence of a homogeneous PNE once $t \geq 2\ell_{\max}$, independent of $N$. Moreover, in our setups, the minimal $t$ is often significantly less than $2\ell_{\max}$ (roughly $0.1 \times (2\ell_{\max})$ to $0.2 \times (2\ell_{\max})$).

2. **Heterogeneous PNE.** Our empirical approach effectively finds heterogeneous PNEs once $N$ exceeds the lower bound given in Thm. 5.7. Nonetheless, because the conditions in Thm. 5.7 are sufficient but not necessary, we observe that a heterogeneous PNE can exist even when $N$ is smaller than that bound. The curve for the heterogeneous PNE is not smooth and exhibits periodic fluctuations. This is due to the fact that the heterogeneous PNE in HD-Game-Probability depends on the PNE in the corresponding HD-Game-Proximity, which itself has a non-smooth dependence on $N$ (Thm. 5.4 and Cor. 5.5).

3. **Coexistence of homogeneous and heterogeneous PNE.** In some games, the heterogeneous PNE curve appears above the homogeneous PNE curve, suggesting that both types may coexist. However, in other cases (e.g., the second row and second column of Fig. 2), no such coexistence is observed. In addition, as $N$ increases, the maximal $t$ required for a heterogeneous PNE tends to be lower than the threshold $\underline{t}$ required for a homogeneous PNE, indicating that the coexistence

of both PNE types becomes increasingly unlikely for large $N$.

# 7. Conclusions

We propose the *Heterogeneous Data Game* to analyze competition among ML models in heterogeneous data markets. By studying PNE under proximity and probability choice models, we derive conditions for the existence of different PNE types, showing key factors that shape competitive ML marketplaces.

# Acknowledgements

Renzhe Xu's research was supported by the National Key R&D Program of China (No. 2023YFA1009500), the National Natural Science Foundation of China (No. 72442024), and the Shanghai Sailing Program (No. 24YF2711600). Bo Li's research was supported by the National Natural Science Foundation of China (Nos. 72171131 and 72133002).

# Impact Statement

This work presents a game-theoretic framework to study competition among ML providers across heterogeneous data sources. By analyzing market equilibrium, it offers insights for designing policies that promote fair and diverse model offerings. Without such safeguards, smaller or less profitable data sources may be neglected, exacerbating inequalities. We aim for our findings to guide policymakers and stakeholders in shaping responsible and equitable ML marketplaces.

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

## A. Omitted Examples

**Example A.1.** Let $N = K = 3$ with $\theta_1 = (0, 0, 1)$, $\theta_2 = (2, 0, 1)$, and $\theta_3 = (0, 1, 1)$. The covariance matrices are $\Sigma_1 = \Sigma_2 = \Sigma_3 = I$, and the weights are given by $\boldsymbol{w} = (w_1, w_2, w_3) = (0.6, 0.25, 0.15)$. A graphical explanation is provided in Fig. 3, where $\theta_1$, $\theta_2$, and $\theta_3$ correspond to the vertices A, B, and C of the triangle, respectively. In this case, the

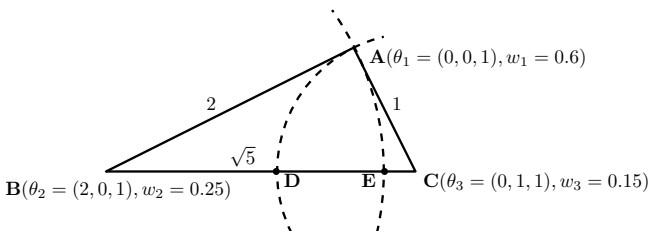

Figure 3. The graphical explanation of Ex. A.1.

Mahalanobis distance reduces to the standard Euclidean distance. It is straightforward to verify that, at a PNE, two players will choose strategy $\theta_1$, while the remaining player will adopt a strategy along the segment DE within the triangle (D and E satisfy that BA = BE and CA = CD), excluding the vertices D and E.

## B. An Approach to Find a Potential Heterogeneous PNE in HD-Game-Probability

### B.1. Approach Design

We first define the following mapping $\mathcal{M}$ from $\Delta_K^N = \underbrace{\Delta_K \times \cdots \times \Delta_K}_{N \text{ times}}$ to $\Delta_K^N$. For a $(\boldsymbol{q}_1, \boldsymbol{q}_2, \ldots, \boldsymbol{q}_N) \in \Delta_K^N$, $\mathcal{M}(\boldsymbol{q}_1, \boldsymbol{q}_2, \ldots, \boldsymbol{q}_N)$ is calculated through Alg. 1.

---
**Algorithm 1** The $\mathcal{M}$ mapping from $\Delta_K^N$ to $\Delta_K^N$
---
1: **Input:** $\boldsymbol{q}_1, \boldsymbol{q}_2, \ldots, \boldsymbol{q}_N \in \Delta_K$
2: $\hat{\theta}_n \leftarrow \bar{\theta}(\boldsymbol{q}_n)$ for all $n \in [N]$
3: $\ell_{n,k} \leftarrow d_M^2(\hat{\theta}_n, \theta_k; \Sigma_k^{-1}) = (\hat{\theta}_n - \theta_k)^\top \Sigma_k (\hat{\theta}_n - \theta_k)$ for all $n \in [N]$ and $k \in [K]$
4: $p_{n,k} \leftarrow \exp(-\ell_{n,k}/t)/(\sum_{i=1}^N \exp(-\ell_{i,k}/t))$ for all $n \in [N]$ and $k \in [K]$
5: $\tilde{q}_{n,k} \leftarrow w_k p_{n,k}(1 - p_{n,k})/(\sum_{j=1}^K w_j p_{n,j}(1 - p_{n,j}))$ for all $n \in [N]$ and $k \in [K]$
6: $\tilde{\boldsymbol{q}}_n \leftarrow (\tilde{q}_{n,1}, \ldots, \tilde{q}_{n,K})$ for all $n \in [N]$
7: **Output:** $(\tilde{\boldsymbol{q}}_1, \tilde{\boldsymbol{q}}_2, \ldots, \tilde{\boldsymbol{q}}_N)$

---

We also need the following definition.

**Definition B.1** ($k_n$). Given a PNE $\hat{\boldsymbol{\theta}}^{\text{Hete}}$ in HD-Game-Proximity, define $k_n$ as follows.

$$k_n \triangleq \left(\text{the index } k \text{ such that } \theta_k = \hat{\theta}_n^{\text{Prox}}\right).$$

Based on the mapping $\mathcal{M}$ and the constant $k_n$, the pseudocode of our approach is provided in Alg. 2. The approach consists of several steps. First, we compute the PNE $\hat{\boldsymbol{\theta}}^{\text{Prox}}$ of the corresponding HD-Game-Proximity using Thm. 5.4 and Cor. 5.5. Then, starting from this strategy profile, we find a fixed point of the mapping $\mathcal{M}$ defined in Alg. 1. Once a fixed point is identified, we output the corresponding strategy profile $\hat{\boldsymbol{\theta}} = (\hat{\theta}_1, \ldots, \hat{\theta}_N)$, where each $\hat{\theta}_n = \bar{\theta}(\boldsymbol{q}_n)$.

### B.2. The Idea of the Approach

The rationality of the approach is based on the following results.

**Lemma B.1.** Let $\boldsymbol{q}^{(1)}, \boldsymbol{q}^{(2)} \in \Delta_K$. Suppose $\|\boldsymbol{q}^{(1)} - \boldsymbol{q}^{(2)}\|_\infty \le \epsilon$. Then, there exists a constant $C > 0$, depending only on $\{\Sigma_k, \theta_k, w_k\}_{k=1}^K$, such that

$$\left\|\bar{\theta}(\boldsymbol{q}^{(1)}) - \bar{\theta}(\boldsymbol{q}^{(2)})\right\|_2 \le C \cdot \epsilon.$$

---

**Algorithm 2** An Approach to Find a Potential Heterogeneous PNE in HD-Game-Probability

1: **Input:** Game parameters $\{\Sigma_k, \theta_k, w_k\}_{k \in [K]}$ and $N$
2: Calculate the PNE $\hat{\theta}^{\text{Prox}}$ based on Thm. 5.4 and Cor. 5.5 in HD-Game-Proximity
3: Calculate $k_n$ for all $n \in [N]$ based on Def. B.1
4: $\boldsymbol{q}_n \leftarrow \underbrace{(0, 0, \ldots, 1, 0, \ldots, 0)}_{\text{the } k_n\text{-th element is 1}}$
5: **while** Not convergent **do**
6: $\quad (\boldsymbol{q}_1, \boldsymbol{q}_2, \ldots, \boldsymbol{q}_N) \leftarrow \mathcal{M}(\boldsymbol{q}_1, \boldsymbol{q}_2, \ldots, \boldsymbol{q}_N)$ given in Alg. 1
7: **end while**
8: $\hat{\theta}_n^{\text{Hete}} \leftarrow \bar{\theta}(\boldsymbol{q}_n)$ for all $n \in [N]$
9: $\hat{\boldsymbol{\theta}}^{\text{Hete}} \leftarrow (\hat{\theta}_1^{\text{Hete}}, \ldots, \hat{\theta}_N^{\text{Hete}})$
10: **Output:** $\hat{\boldsymbol{\theta}}^{\text{Hete}}$

---

Let $\bar{C}$ be the constant that only depends on $\{\Sigma_k, \theta_k, w_k\}_{k=1}^K$ in Lem. B.1. For any $t > 0$ and $\beta \geq 1$, define the space

$$\mathcal{Q}^{(t,\beta)} \triangleq \mathcal{Q}_1^{(t,\beta)} \times \mathcal{Q}_2^{(t,\beta)} \times \cdots \times \mathcal{Q}_N^{(t,\beta)} \tag{13}$$

where $\forall n \in [N]$,

$$\mathcal{Q}_n^{(t,\beta)} \triangleq \left\{ \boldsymbol{q} \in \Delta_K : \left\| \boldsymbol{q} - \underbrace{(0, 0, \ldots, 1, 0, \ldots, 0)}_{\text{the } k_n\text{-th element is 1}} \right\|_\infty \leq t^\beta / \bar{C} \right\}.$$

Based on the above definition, we could provide two important properties of the mapping $\mathcal{M}$.

**Lemma B.2.** *For any* $(\boldsymbol{q}_1, \ldots, \boldsymbol{q}_N) \in \Delta_K^N$, *let* $\hat{\boldsymbol{\theta}} = (\hat{\theta}_1, \hat{\theta}_2, \ldots, \hat{\theta}_N)$ *where* $\hat{\theta}_n = \bar{\theta}(\boldsymbol{q}_n)$, $\forall n \in [N]$. *If* $(\boldsymbol{q}_1, \ldots, \boldsymbol{q}_N)$ *is a fixed point of the mapping* $\mathcal{M}$, *then for all* $n \in [N]$,

$$\left. \frac{\partial u_n(\theta, \hat{\boldsymbol{\theta}}_{-n})}{\partial \theta} \right|_{\theta = \hat{\theta}_n} = 0.$$

**Lemma B.3.** *When* $\beta > 1$, *there exists a constant* $\underline{t}$, *depending only on* $\{\Sigma_k, \theta_k, w_k\}_{k=1}^K$ *and* $\beta$, *such that when* $t \leq \underline{t}$, *for any* $(\boldsymbol{q}_1, \boldsymbol{q}_2, \ldots, \boldsymbol{q}_N) \in \mathcal{Q}^{(t,\beta)}$ *defined in Eq.* (13), *then* $\mathcal{M}(\boldsymbol{q}_1, \boldsymbol{q}_2, \ldots, \boldsymbol{q}_N) \in \mathcal{Q}^{(t,\beta)}$.

Based on the Brouwer fixed-point theorem, Lem. B.3 guarantees the existence of a fixed point $(\boldsymbol{q}_1, \ldots, \boldsymbol{q}_N) \in \mathcal{Q}^{(t,\beta)}$ for the mapping $\mathcal{M}$. Consequently, by Lem. B.2, the corresponding strategy profile $\hat{\boldsymbol{\theta}}^{\text{Hete}} = (\hat{\theta}_1^{\text{Hete}}, \ldots, \hat{\theta}_N^{\text{Hete}})$, where each $\hat{\boldsymbol{\theta}}_n^{\text{Hete}} = \bar{\theta}(\boldsymbol{q}_n)$, satisfies the zero partial gradient condition at $\hat{\theta}_n^{\text{Hete}}$ for $u_n(\theta, \hat{\boldsymbol{\theta}}_{-n}^{\text{Hete}})$ for all $n \in [N]$, which is a necessary condition for a PNE. Therefore, $\hat{\boldsymbol{\theta}}^{\text{Hete}}$ serves as a candidate for a heterogeneous PNE in HD-Game-Probability.

The complete proof is available in the full version of the paper at `https://arxiv.org/abs/2505.07688`.

## C. Omitted Proofs

**Additional notations.** For any two vectors $\boldsymbol{x} = (x_1, \ldots, x_M) \in \mathbb{R}^M$ and $\boldsymbol{y} = (y_1, \ldots, y_M) \in \mathbb{R}^M$, we say $\boldsymbol{x}$ is dominated by $\boldsymbol{y}$ if for all $i \in [M]$, $y_i \leq x_i$ and there exists a $j \in [M]$, $y_j < x_j$.

### C.1. Proof of Prop. 4.1

We first prove Eq. (7).

*Proof of Eq.* (7). According to the definition of the Mahalanobis distance, we have

$$\bar{\theta}(\boldsymbol{q}) = \arg\min_\theta \sum_{k=1}^K q_k d_M^2(\theta, \theta_k; \Sigma_k^{-1}) = \arg\min_\theta \sum_{k=1}^K q_k (\theta - \theta_k)^T \Sigma_k (\theta - \theta_k)$$

Denote the part in $\arg\min$ in the right-hand side of the above equation as $L(\theta)$. And since $q \in \Delta_K$ and $\Sigma_k \succ 0$, we have

$$\nabla L(\theta) = 2\sum_{k=1}^{K} q_k \Sigma_k (\theta - \theta_k), \quad \nabla^2 L(\theta) = 2\sum_{k=1}^{K} q_k \Sigma_k \succ 0.$$

As a result, $L(\theta)$ is strictly convex and it has the unique minimizer $\bar{\theta}(q)$. In addition, $\bar{\theta}(q)$ must satisfy that $\nabla L(\bar{\theta}(q)) = 0$, which means

$$2\sum_{k=1}^{K} q_k \Sigma_k (\bar{\theta}(q) - \theta_k) = 0.$$

As a result,

$$\bar{\theta}(q) = \left(\sum_{k=1}^{K} q_k \Sigma_k\right)^{-1} \left(\sum_{k=1}^{K} q_k \Sigma_k \theta_k\right).$$

Now the claim follows. $\qquad\square$

For the proof of Prop. 4.1, we first introduce the following lemma.

**Lemma C.1.** *Let $\vartheta$ be defined in Eq. (6). Then for any $\theta \notin \vartheta$, there exists a $\theta^* \in \vartheta$ such that for all $k \in [K]$, $d_M(\theta^*, \theta_k; \Sigma_k^{-1}) < d_M(\theta, \theta_k; \Sigma_k^{-1})$.*

*Proof of Lem. C.1.* Define the following function $f : \mathbb{R}^D \to \mathbb{R}^K$ as

$$f(\theta) = \left(d_M^2(\theta, \theta_1; \Sigma_1^{-1}), d_M^2(\theta, \theta_2; \Sigma_2^{-1}), \ldots, d_M^2(\theta, \theta_K; \Sigma_K^{-1})\right).$$

For any $k \in [K]$, define $f_k(\theta) = d_M^2(\theta, \theta_k; \Sigma_k^{-1})$. Note that for any $k \in [K]$, since $\Sigma_k \succ 0$, we have $d_M^2(\theta, \theta_k; \Sigma_k^{-1}) = (\theta - \theta_k)^\top \Sigma_k (\theta - \theta_k)$ is convex w.r.t. $\theta$. As a result, according to Boyd (2004), for every Pareto optimal point $\theta$ of $f$, there is some $q \in \Delta_K$ such that

$$\theta = \arg\min_{\theta'} \sum_{k=1}^{K} q_k d_M^2(\theta', \theta_k; \Sigma_k^{-1}) = \bar{\theta}(q).$$

Hence, the set $\vartheta$ defined in Eq. (6) contains all Pareto optimal points.

Furthermore, for any points $\theta \notin \vartheta$, there must exist a $\theta' \in \vartheta$ such that $f(\theta')$ dominates $f(\theta)$ (Ehrgott, 2005), which proves the claim. $\qquad\square$

Now we could prove Prop. 4.1.

*proof of Prop. 4.1.* (1) In the proximity model, let $\hat{\theta} = (\hat{\theta}_1, \hat{\theta}_2, \ldots, \hat{\theta}_K)$ be a PNE. If $\hat{\theta}_k \in \vartheta, \forall k \in [K]$, then the claim is already satisfied. Now suppose there exists an index $n$ such that $\hat{\theta}_n \notin \vartheta$. According to Lem. C.1, there exists a policy $\theta' \in \vartheta$ such that $\forall k \in [K], \ell'_{n,k} = d_M^2(\theta', \theta_k; \Sigma_k^{-1}) < d_M^2(\hat{\theta}_n, \theta_k; \Sigma_k^{-1}) = \ell_{n,k}$. Consider the new strategy profile $\tilde{\theta} = (\theta', \hat{\theta}_{-n})$. We now show that $\tilde{\theta}$ is also a PNE.

Since $g_n^{\text{PROX}}$ is decreasing on the $n$-th element and $\hat{\theta}$ is a PNE, we have that

$$u_n(\hat{\theta}) = \sum_{k=1}^{K} w_k g_n^{\text{PROX}}(\ell_{1,k}, \ldots, \ell_{N,k}) \leq \sum_{k=1}^{K} w_k g_n^{\text{PROX}}(\ell_{1,k}, \ldots, \ell_{n-1,k}, \ell'_{n,k}, \ell_{n+1,k}, \ldots, \ell_{N,k}) = u_n(\tilde{\theta}) \leq u_n(\hat{\theta}).$$

As a result, $u_n(\tilde{\theta}) = u_n(\hat{\theta})$ and player $n$ will not benefit by deviation. Furthermore, $\forall k \in [K]$, we have $g_n^{\text{PROX}}(\ell_{1,k}, \ldots, \ell_{N,k}) = g_n^{\text{PROX}}(\ell_{1,k}, \ldots, \ell_{n-1,k}, \ell'_{n,k}, \ell_{n+1,k}, \ldots, \ell_{N,k})$. (Otherwise, player $n$ would benefit by deviation.) Since $\ell'_{n,k} < \ell_{n,k}$, we have

$$g_n^{\text{PROX}}(\ell_{1,k}, \ldots, \ell_{N,k}) = g_n^{\text{PROX}}(\ell_{1,k}, \ldots, \ell_{n-1,k}, \ell'_{n,k}, \ell_{n+1,k}, \ldots, \ell_{N,k}) = 0$$

and hence $\ell_{n,k} > \ell'_{n,k} > \min_{i \in [N]} \ell_{i,k}, \forall k \in [K]$. As a result, this deviation will not affect any other players' utility. Specifically, consider any other player $j \in [N] \backslash \{n\}$. We have

$$g_j^{\text{PROX}}(\ell_{1,k}, \ldots, \ell_{n-1,k}, \ell'_{n,k}, \ell_{n+1,k}, \ldots, \ell_{N,k}) = g_j^{\text{PROX}}(\ell_{1,k}, \ldots, \ell_{N,k}). \tag{14}$$

Furthermore, since $\ell'_{n,k} < \ell_{n,k}$, we have that for any $\theta'' \in \mathbb{R}^D$ and $\ell''_{j,k} = d_M^2(\theta'', \theta_k; \Sigma_k^{-1})$, we have $\forall j \in [N]$,

$$
\begin{aligned}
&g_j^{\text{PROX}}(\ell_{1,k}, \ldots, \ell_{j-1,k}, \ell''_{j,k}, \ell_{j+1,k}, \ldots, \ell_{n-1,k}, \ell'_{n,k}, \ell_{n+1,k}, \ldots, \ell_{N,k}) \\
&\leq g_j^{\text{PROX}}(\ell_{1,k}, \ldots, \ell_{j-1,k}, \ell''_{j,k}, \ell_{j+1,k}, \ldots, \ell_{N,k}).
\end{aligned}
\tag{15}
$$

As a result,

$$
\begin{aligned}
&u_j((\theta'', \tilde{\boldsymbol{\theta}}_{-j})) \\
&= \sum_{k=1}^{K} w_k \cdot g_j^{\text{PROX}}(\ell_{1,k}, \ldots, \ell_{j-1,k}, \ell''_{j,k}, \ell_{j+1,k}, \ldots, \ell_{n-1,k}, \ell'_{n,k}, \ell_{n+1,k}, \ldots, \ell_{N,k}) \\
&\leq \sum_{k=1}^{K} w_k \cdot g_j^{\text{PROX}}(\ell_{1,k}, \ldots, \ell_{j-1,k}, \ell''_{j,k}, \ell_{j+1,k}, \ldots, \ell_{N,k}) && \text{(by Eq. (15))} \\
&\leq \sum_{k=1}^{K} w_k \cdot g_j^{\text{PROX}}(\ell_{1,k}, \ldots, \ell_{j-1,k}, \ell_{j,k}, \ell_{j+1,k}, \ldots, \ell_{N,k}) && (\hat{\boldsymbol{\theta}} \text{ is a PNE and hence } u_j(\hat{\boldsymbol{\theta}}) \geq u_j(\theta'', \hat{\boldsymbol{\theta}}_{-j})) \\
&= \sum_{k=1}^{K} w_k \cdot g_j^{\text{PROX}}(\ell_{1,k}, \ldots, \ell_{n-1,k}, \ell'_{n,k}, \ell_{n+1,k}, \ldots, \ell_{N,k}) && \text{(by Eq. (14))} \\
&= u_j(\tilde{\boldsymbol{\theta}}).
\end{aligned}
$$

Therefore, any player will not benefit by deviation from $\tilde{\boldsymbol{\theta}}$ and $\tilde{\boldsymbol{\theta}}$ is a PNE.

To prove the original claim, we can keep this procedure. After at most $N$ steps, we can get a new PNE $\boldsymbol{\theta}^* = (\theta_1^*, \ldots, \theta_N^*)$ from $\boldsymbol{\theta}$ and $\theta_k^* \in \vartheta, \forall k \in [K]$. Now the claim follows.

(2) In the probability model, suppose there exists a PNE $\boldsymbol{\theta} = (\theta_1, \theta_2, \ldots, \theta_N)$ and an index $n \in [N]$ such that $\theta_n \notin \vartheta$. Then according to Lem. C.1, there exists $\theta' \in \vartheta$ such that $\forall k \in [K], d_M(\theta', \theta_k; \Sigma_k^{-1}) < d_M(\theta_n, \theta_k; \Sigma_k^{-1})$. Since $g_n^{\text{PROP}}$ is strictly decreasing on the $n$-th element, we could conclude that player $n$ will benefit if deviating to the policy $\theta'$, which leads to a contradiction. Now the claim follows. $\square$

## C.2. Proof of Thm. 5.1

We first introduce the following lemma.

**Lemma C.2.** *Suppose that Assump. 4.1 holds. Let $\boldsymbol{q} \in \Delta_K$ and $\theta = \bar{\theta}(\boldsymbol{q})$. Then for any $k_0 \in [K]$ such that $q_{k_0} > 0$, there exists a strategy $\tilde{\theta} \in \vartheta$ such that*

$$\forall k \in [K] \backslash \{k_0\}, \quad d_M(\tilde{\theta}, \theta_k; \Sigma_k^{-1}) < d_M(\theta, \theta_k; \Sigma_k^{-1}).$$

*Proof.* For any $k \in [K] \backslash \{k_0\}$, define $v_k = \Sigma_k(\theta - \theta_k)$. Furthermore, define the following matrix

$$A = \begin{pmatrix} v_1 & v_2 & \cdots & v_{k_0-1} & v_{k_0+1} & \cdots v_{K.} \end{pmatrix}$$

Then for any $\boldsymbol{y} \in \mathbb{R}^{K-1}$ such that $\boldsymbol{y} \geq 0$ and $\boldsymbol{y} \neq 0$, we must have $A\boldsymbol{y} \neq 0$. Otherwise, we can construct a new $\boldsymbol{q}'$ such that

$$q_k' = \begin{cases} y_k / \left( \sum_{k'=1}^{K-1} y_{k'} \right) & \text{if } k < k_0, \\ 0 & \text{if } k = k_0, \\ y_{k-1} / \left( \sum_{k'=1}^{K-1} y_{k'} \right) & \text{if } k > k_0. \end{cases}$$

As a result,

$$\sum_{k=1}^{K} q'_k \Sigma_k (\theta - \theta_k) = \sum_{k=1}^{K} q'_k v_k = A\boldsymbol{y}/\left(\sum_{k'=1}^{K-1} y_{k'}\right) = 0$$

and $\theta = \bar{\theta}(\boldsymbol{q}') = \bar{\theta}(\boldsymbol{q})$, which violates Assump. 4.1.

According to Gordan's theorem (Lem. D.1, (Mangasarian, 1994)), there must exist a vector $x \in \mathbb{R}^D$ such that $(-A)^\top x > 0$, which means for all $k \in [K]\backslash\{k_0\}$, we have $v_k^\top x < 0$. Now construct the following $\theta'_t = \theta + t \cdot x$ with any $t \geq 0$. We can get that, for any $k \in [K]\backslash\{k_0\}$,

$$\left.\frac{\mathrm{d}d_M^2(\theta'_t, \theta_k; \Sigma_k^{-1})}{\mathrm{d}t}\right|_{t=0} = \left.\frac{\mathrm{d}\left(\theta + t \cdot x - \theta_k\right)^\top \Sigma_k \left(\theta + t \cdot x - \theta_k\right)}{\mathrm{d}t}\right|_{t=0} = \left.2x^\top \Sigma_k (\theta + t \cdot x - \theta_k)\right|_{t=0}$$

$$= 2x^\top \Sigma_k (\theta - \theta_k) = 2x^\top v_k < 0.$$

As a result, there must exist a $t_k > 0$ such that for any $0 < t < t_k$, we have $d_M^2(\theta'_t, \theta_k; \Sigma_k^{-1}) < d_M^2(\theta, \theta_k; \Sigma_k^{-1})$. Now choose $t' = \min\{t_1, t_2, \ldots, t_{k_0-1}, t_{k_0+1}, \ldots, t_K\}/2$ and let $\theta' = \theta + t' \cdot x$. Then for all $k \in [K]\backslash\{k_0\}$, we must have $d_M(\theta', \theta_k; \Sigma_k^{-1}) < d_M(\theta, \theta_k; \Sigma_k^{-1})$.

Now if $\theta' \in \vartheta$, the claim has already follows. When $\theta' \notin \vartheta$, according to the proof of Lem. C.1 (see App. C.1), $\vartheta$ contains all Pareto optimal points. As a result, there must exist a strategy $\tilde{\theta} \in \vartheta$ such that for all $k \in [K]\backslash\{k_0\}, d_M(\tilde{\theta}, \theta_k; \Sigma_k^{-1}) < d_M(\theta', \theta_k; \Sigma_k^{-1}) < d_M(\theta, \theta_k; \Sigma_k^{-1})$. Now the claim follows. $\qquad\square$

Then we could prove Thm. 5.1.

*Proof of Thm. 5.1.* (1) We first consider the case where $w_1 < 0.5$. Suppose a PNE $\hat{\boldsymbol{\theta}} = (\hat{\theta}_1, \hat{\theta}_2)$ exists. According to Prop. 4.1, we can assume that $\hat{\theta}_1, \hat{\theta}_2 \in \vartheta$. Since the sum of the utilities of two players is 1, there must exist one player that has utility not greater than 0.5. Without loss of generality, we assume that $u_1(\hat{\boldsymbol{\theta}}) \leq 0.5$ and $u_2(\hat{\boldsymbol{\theta}}) \geq 0.5$. Now we construct a new policy $\theta'$ for player 1 such that $u_1(\theta', \hat{\theta}_2) > 0.5$.

According to Assump. 4.1, there exists a unique $\boldsymbol{q} \in \Delta_K$ such that $\hat{\theta}_2 = \bar{\theta}(\boldsymbol{q})$. Since $\boldsymbol{q} \in \Delta_K$, there must exist one element $k_0$ such that $q_{k_0} > 0$. According to Lem. C.2, there must exist a $\theta' \in \vartheta$ such that

$$\forall k \in [K]\backslash\{k_0\}, \quad d_M(\theta', \theta_k; \Sigma_k^{-1}) < d_M(\theta, \theta_k; \Sigma_k^{-1}).$$

Now by the proximity model as shown in Eq. (2), we have that

$$u_1(\theta', \hat{\theta}_2) \geq \sum_{k \in [K]\backslash\{k_0\}} w_k = 1 - w_{k_0} \geq 1 - w_1 > 1 - 0.5 = 0.5 \geq u_1(\hat{\boldsymbol{\theta}}).$$

As a result, $\hat{\boldsymbol{\theta}}$ is not a PNE, which leads to a contradiction.

(2) We then consider the case when $w_1 \geq 0.5$.

We first show that $\hat{\boldsymbol{\theta}} = (\theta_1, \theta_1)$ is a PNE. In this strategy profile, Since two players choose the same strategy $\theta_1$, we have $u_1(\hat{\boldsymbol{\theta}}) = u_2(\hat{\boldsymbol{\theta}}) = 0.5$. In addition, when any player deviate from the strategy $\theta_1$, he could have higher loss on data source 1 and hence could have utility at most $\sum_{k=2}^{K} w_k = 1 - w_1 \leq 0.5$. As a result, $\hat{\boldsymbol{\theta}} = (\theta_1, \theta_1)$ is a PNE.

Furthermore, consider the case when $w_1 > 0.5$. Suppose there exists another PNE $\hat{\boldsymbol{\theta}}' = (\hat{\theta}_1, \hat{\theta}_2) \neq \hat{\boldsymbol{\theta}}$. We consider the following two cases.

1. Suppose $\hat{\theta}_1, \hat{\theta}_2 \neq \theta_1$. Since $u_1(\hat{\boldsymbol{\theta}}') + u_2(\hat{\boldsymbol{\theta}}') = 1$, there exist one player such that his utility is not greater than 0.5. Without loss of generality, we assume $u_1(\hat{\boldsymbol{\theta}}') \leq 0.5$. Then if player 1 choose strategy $\theta_1$, he will get utility at least $w_1 > 0.5 \geq u_1(\hat{\boldsymbol{\theta}}')$, which leads to a contradiction.

2. Suppose one player choose $\theta_1$ and the other player does not. Without loss of generality, we assume $\hat{\theta}'_1 = \theta_1$ and $\hat{\theta}'_2 \neq \theta_1$. In this case, $u_2(\hat{\boldsymbol{\theta}}') \leq \sum_{k=2}^{K} w_k = 1 - w_1 < 0.5$. However, if player 2 choose strategy $\theta_1$, he will have the same strategy with player 1 and get utility $0.5 > u_2(\hat{\boldsymbol{\theta}}')$, which leads to a contraction.

To conclude, $\hat{\boldsymbol{\theta}} = (\theta_1, \theta_1)$ is the unique PNE when $w_1 > 0.5$.

Now the claim follows. $\qquad\qquad\qquad\qquad\qquad\qquad\qquad\qquad\qquad\qquad\qquad\qquad\qquad\qquad\qquad\quad\square$

## C.3. Proof of Thm. 5.2

*Proof.* (1) We first show that, if a PNE exists, the only possible PNE is that both players choose $\bar{\theta}(\boldsymbol{w})$.

Suppose that $\hat{\boldsymbol{\theta}} = (\hat{\theta}_1, \hat{\theta}_2)$ is a PNE. According to the definition of PNE, we must have

$$\hat{\theta}_1 \in \arg\max_\theta u_1(\theta, \hat{\theta}_2) = \arg\max_\theta \sum_{k=1}^K w_k \cdot p_{1,k}(\theta)$$

where

$$p_{1,k}(\theta) = \frac{\exp\left(-(\theta - \theta_k)^\top \Sigma_k (\theta - \theta_k)/t\right)}{\exp\left(-(\theta - \theta_k)^\top \Sigma_k (\theta - \theta_k)/t\right) + \exp\left(-\left(\hat{\theta}_2 - \theta_k\right)^\top \Sigma_k \left(\hat{\theta}_2 - \theta_k\right)/t\right)}.$$

Note that

$$\frac{\partial p_{1,k}(\theta)}{\partial \theta}$$

$$= \frac{\exp\left(-(\theta - \theta_k)^\top \Sigma_k (\theta - \theta_k)/t\right) \cdot \exp\left(-\left(\hat{\theta}_2 - \theta_k\right)^\top \Sigma_k \left(\hat{\theta}_2 - \theta_k\right)/t\right)}{\left(\exp\left(-(\theta - \theta_k)^\top \Sigma_k (\theta - \theta_k)/t\right) + \exp\left(-\left(\hat{\theta}_2 - \theta_k\right)^\top \Sigma_k \left(\hat{\theta}_2 - \theta_k\right)/t\right)\right)^2} \cdot \left(-\frac{2}{t} \cdot \Sigma_k (\theta - \theta_k)\right)$$

$$= -\frac{2}{t} \cdot p_{1,k}(\theta)(1 - p_{1,k}(\theta))\Sigma_k(\theta - \theta_k).$$

As a result,

$$\frac{\partial u_1(\theta, \hat{\theta}_2)}{\partial \theta} = \sum_{k=1}^K w_k \cdot \frac{\partial p_{1,k}(\theta)}{\partial \theta} = -\frac{2}{t} \cdot \sum_{k=1}^K w_k p_{1,k}(\theta)(1 - p_{1,k}(\theta))\Sigma_k(\theta - \theta_k).$$

Hence,

$$\left.\frac{\partial u_1(\theta, \hat{\theta}_2)}{\partial \theta}\right|_{\theta=\hat{\theta}_1} = -\frac{2}{t} \cdot \sum_{k=1}^K w_k p_{1,k}(\hat{\theta}_1)(1 - p_{1,k}(\hat{\theta}_1))\Sigma_k(\hat{\theta}_1 - \theta_k) = 0. \qquad (16)$$

Similarly, we can get that

$$\left.\frac{\partial u_2(\hat{\theta}_1, \theta)}{\partial \theta}\right|_{\theta=\hat{\theta}_2} = -\frac{2}{t} \cdot \sum_{k=1}^K w_k p_{2,k}(\hat{\theta}_2)(1 - p_{2,k}(\hat{\theta}_2))\Sigma_k(\hat{\theta}_2 - \theta_k) = 0.$$

where

$$p_{2,k}(\theta) = \frac{\exp\left(-(\theta - \theta_k)^\top \Sigma_k (\theta - \theta_k)/t\right)}{\exp\left(-\left(\hat{\theta}_1 - \theta_k\right)^\top \Sigma_k \left(\hat{\theta}_1 - \theta_k\right)/t\right) + \exp\left(-(\theta - \theta_k)^\top \Sigma_k (\theta - \theta_k)/t\right)}.$$

Note that $p_{1,k}(\hat{\theta}_1) + p_{2,k}(\hat{\theta}_2) = 1$. As a result,

$$\left.\frac{\partial u_2(\theta, \hat{\theta}_2)}{\partial \theta}\right|_{\theta=\hat{\theta}_1} = -\frac{2}{t} \cdot \sum_{k=1}^K w_k p_{1,k}(\hat{\theta}_1)(1 - p_{1,k}(\hat{\theta}_1))\Sigma_k(\hat{\theta}_1 - \theta_k) = -\frac{2}{t} \cdot \sum_{k=1}^K w_k p_{1,k}(\hat{\theta}_1)p_{2,k}(\hat{\theta}_2)\Sigma_k(\hat{\theta}_1 - \theta_k) = 0.$$

$$\left.\frac{\partial u_2(\hat{\theta}_1, \theta)}{\partial \theta}\right|_{\theta=\hat{\theta}_2} = -\frac{2}{t} \cdot \sum_{k=1}^K w_k p_{2,k}(\hat{\theta}_2)(1 - p_{2,k}(\hat{\theta}_2))\Sigma_k(\hat{\theta}_2 - \theta_k) = -\frac{2}{t} \cdot \sum_{k=1}^K w_k p_{1,k}(\hat{\theta}_1)p_{2,k}(\hat{\theta}_2)\Sigma_k(\hat{\theta}_2 - \theta_k) = 0.$$

Hence,

$$\sum_{k=1}^K w_k p_{1,k}(\hat{\theta}_1)p_{2,k}(\hat{\theta}_2)\Sigma_k(\hat{\theta}_1 - \theta_k) = 0 = \sum_{k=1}^K w_k p_{1,k}(\hat{\theta}_1)p_{2,k}(\hat{\theta}_2)\Sigma_k(\hat{\theta}_2 - \theta_k)$$

and therefore

$$\sum_{k=1}^{K} w_k p_{1,k}(\hat{\theta}_1) p_{2,k}(\hat{\theta}_2) \Sigma_k (\hat{\theta}_1 - \hat{\theta}_2) = 0.$$

Define matrix $A = \sum_{k=1}^{K} w_k p_{1,k}(\hat{\theta}_1) p_{2,k}(\hat{\theta}_2) \Sigma_k$ and we have $A(\hat{\theta}_1 - \hat{\theta}_2) = 0$. Note that for all $k \in [K]$, $w_k, p_{1,k}(\hat{\theta}_1), p_{2,k}(\hat{\theta}_2) > 0$ and $\Sigma_k \succ 0$. As a result, $A \succ 0$ and we must have $\hat{\theta}_1 = \hat{\theta}_2$. Note that when $\hat{\theta}_1 = \hat{\theta}_2$, $p_{1,k}(\hat{\theta}_1) = p_{2,k}(\hat{\theta}_2) = 1/2$. Now Eq. (16) becomes

$$-\frac{2}{t} \cdot \sum_{k=1}^{K} w_k \cdot \frac{1}{2} \cdot \frac{1}{2} \Sigma_k (\hat{\theta}_1 - \theta_k) = 0.$$

As a result, $\hat{\theta}_1 = \bar{\theta}(\boldsymbol{w}) = \hat{\theta}_2$. Hence, if a PNE exists, the only possible PNE is that both players choose $\bar{\theta}(\boldsymbol{w})$.

The claim then follows from the proof of the more general result given by Thm. 5.6 (see Lems. C.3 and C.4 in App. C.6 for details).

$\square$

## C.4. Proof of Prop. 5.3

*Proof.* Suppose there exists a PNE $\hat{\boldsymbol{\theta}} = (\hat{\theta}_1, \hat{\theta}_2, \ldots, \hat{\theta}_N)$ such that there are two players choose the same strategy and the strategy is outside the set $\{\theta_1, \theta_2, \ldots, \theta_K\}$. Without loss of generality, we assume that $\hat{\theta}_1 = \hat{\theta}_2 \notin \{\theta_1, \ldots, \theta_K\}$.

Define $\mathcal{K}$ as the set of data sources that player 1 and 2 could get positive utility, i.e.,

$$\mathcal{K} \triangleq \{k : \forall j \in [N], d_M(\hat{\theta}_1, \theta_k; \Sigma_k^{-1}) \leq d_M(\hat{\theta}_j, \theta_k; \Sigma_k^{-1})\}.$$

Note that $\mathcal{K}$ cannot be an empty set, as player 1 could otherwise deviate to $\theta_1$ and achieve a positive and higher utility. For any $k \in \mathcal{K}$, let

$$k_n = \left| \{j : d_M(\hat{\theta}_j, \theta_k; \Sigma_k^{-1}) = d_M(\hat{\theta}_1, \theta_k; \Sigma_k^{-1})\} \right|$$

be the number of players that achieve the minimal loss in data source $k$ in the PNE $\hat{\boldsymbol{\theta}}$. Note that $k_n \geq 2$ since $\hat{\theta}_1 = \hat{\theta}_2$. Then we can get that

$$u_1(\hat{\boldsymbol{\theta}}) = u_2(\hat{\boldsymbol{\theta}}) = \sum_{k \in \mathcal{K}} \frac{w_k}{k_n}.$$

We consider two cases about $\mathcal{K}$.

1. Consider the case when $|\mathcal{K}| = 1$ and $\mathcal{K} = \{k_0\}$. If player 1 deviates to policy $\theta_{k_0}$, he will become the only player that has the smallest loss on data source $k$ and hence have a utility at least $w_{k_0} > w_{k_0}/n_{k_0}$, which leads to a contradiction.

2. Consider the case when $|\mathcal{K}| \geq 2$. We further consider two cases about $\hat{\theta}_1$.

   (a) Consider the case when $\hat{\theta}_1 \notin \vartheta$. Then according to the proof of Lem. C.1 (see App. C.1), there must exist a $\theta \in \vartheta$ such that for all $k \in [K]$, $d_M(\theta, \theta_k; \Sigma_k^{-1}) < d_M(\hat{\theta}_2, \theta_k; \Sigma_k^{-1})$. As a result, if player 1 deviates to the policy $\theta$, he will get utility at least $\sum_{k \in \mathcal{K}} w_k > \sum_{k \in \mathcal{K}} w_k/k_n = u_1(\hat{\boldsymbol{\theta}})$, which leads to a contradiction.

   (b) Consider the case when $\hat{\theta}_1 \in \vartheta$. Let $k_0$ be the smallest element in $\mathcal{K}$. According to Assump. 4.1, let $\boldsymbol{q} \in \Delta_K$ be the unique vector such that $\hat{\theta}_1 = \bar{\theta}(\boldsymbol{q})$. Since $\hat{\theta} \neq \theta_{k_0}$ by assumption, there must exist a $k_1 \in [K] \backslash \{k_0\}$ such that $q_{k_1} > 0$. Now according to Lem. C.2, there exists a strategy $\theta \in \vartheta$ such that for all $k \in [K] \backslash \{k_1\}$, we have $d_M(\theta, \theta_k; \Sigma_k^{-1}) < d_M(\hat{\theta}_2, \theta_k; \Sigma_k^{-1})$. Let $k_2$ be the second smallest element in $\mathcal{K}$. As a result,

   $$u_1(\theta, \hat{\boldsymbol{\theta}}_{-1}) = \sum_{k \in \mathcal{K} \backslash \{k_1\}} w_k \geq \sum_{k \in \mathcal{K} \backslash \{k_2\}} w_k > \frac{w_{k_0}}{2} + \frac{w_{k_2}}{2} + \sum_{k \in \mathcal{K} \backslash \{k_0, k_2\}} \frac{w_k}{k_n} \geq u_1(\hat{\boldsymbol{\theta}}).$$

   Therefore, player 1 will have a higher utility if deviating to the policy $\theta$, which leads to a contradiction.

To conclude, all cases lead to a contradiction. As a result, the claim follows. $\square$

## C.5. Proof of Thm. 5.4

*Proof.* (1) We first show the existence of PNE under the condition in Eq. (10).

Note that $z^*$ exists since $h(z) \to \infty$ when $z \to 0$ and $h(z) = 0$ when $z > 1$. Moreover, since $h(z)$ is right continuous, it must hold that $h(z^*) \geq N$. Under the condition in Eq. (10), we have

$$h\left(\frac{w'_{k_0}}{3}\right) = \sum_{k=1}^{k_0} \left\lfloor \frac{3w'_k}{w'_{k_0}} \right\rfloor \leq N.$$

In addition, for any $\epsilon > 0$, we have

$$h\left(\frac{w'_{k_0}}{3} + \epsilon\right) = \sum_{k=1}^{k_0} \left\lfloor \frac{3w'_k}{w'_{k_0} + 3\epsilon} \right\rfloor \leq \left(\sum_{k=1}^{k_0-1} \left\lfloor \frac{3w'_k}{w'_{k_0} + 3\epsilon} \right\rfloor\right) + 2 < \left(\sum_{k=1}^{k_0-1} \left\lfloor \frac{3w'_k}{w'_{k_0}} \right\rfloor\right) + \left\lfloor \frac{3w'_{k_0}}{w'_{k_0}} \right\rfloor \leq N.$$

Hence, it must hold that $z^* \leq w'_{k_0}/3$. Then define

$$\forall k \in [k_0], \quad m'_k = \left\lfloor \frac{w'_k}{z^*} \right\rfloor.$$

As a result, we have that $m'_k \geq m'_{k_0} = \lfloor w'_{k_0}/z^* \rfloor \geq 3$ for all $k \in [k_0]$. In addition, due to Eq. (10), by making $\epsilon > 0$ small enough, we have that

$$h\left(\sum_{j=k_0+1}^{K} w_j + \epsilon\right) = \sum_{k=1}^{k_0} \left\lfloor \frac{w'_k}{\left(\sum_{j=k_0+1}^{K} w_j\right) + \epsilon} \right\rfloor \geq \sum_{k=1}^{k_0} \left(\left\lceil \frac{w'_k}{\left(\sum_{j=k_0+1}^{K} w_j\right)} \right\rceil - 1\right) \geq N.$$

We have that $z^* > \sum_{j=k_0+1}^{K} w_j$.

We construct the PNE based on two cases.

1. Consider the scenario when $h(z^*) = \sum_{k=1}^{k_0} m'_k = N$. Then let $m^*_k = m'_k$ for all $k \in [k_0]$.

2. Consider the scenario when $h(z^*) = \sum_{k=1}^{k_0} m'_k > N$. Note that by the choice of $z^*$, $h(z + \epsilon) < N$ for all $\epsilon > 0$. Define the set $\mathcal{K} = \{k \in [k_0] : w'_k/z^* = m'_k\}$. As a result, when $\epsilon \to 0$, $h(z^* + \epsilon) = h(z^*) - |\mathcal{K}| < N$. Hence, it must hold that $|\mathcal{K}| > h(z^*) - N$. Let $\mathcal{K}'$ be the set of the $(h(z^*) - N)$ smallest elements in $\mathcal{K}$. Define $m^*_k$ as follows.

$$\forall k \in [k_0], \quad m^*_k = \begin{cases} m'_k & \text{if } k \notin \mathcal{K}' \\ m'_k - 1 & \text{if } k \in \mathcal{K}'. \end{cases} \tag{17}$$

Now it holds that $\sum_{k=1}^{k_0} m^*_k = N$.

Construct a strategy profile $\hat{\boldsymbol{\theta}}^* = (\hat{\theta}_1, \hat{\theta}_2, \ldots, \hat{\theta}_N)$ such that $m^*_1$ players choose strategy $\theta_1$, $m^*_2$ players choose strategy $\theta_2$, ..., and $m^*_{k_0}$ players choose strategy $\theta_{k_0}$. In this profile, for any player that chooses strategy $\theta_k$, by the construction of $w'_k$ in Eq. (11), he will get utility at least $w'_k/m^*_k$. Moreover, by the construction of $m^*_k$ and $m'_k$, we have that $w'_k/m^*_k \geq w'_k/m'_k \geq z^*$. Hence, all players have a utility at least $z^*$. Then we show that for any player $i$ that chooses strategy $\theta_k$ with $k \in [k_0]$, he could only get utility at most $z^*$ by deviation. Consider the two cases of the deviated strategy $\theta'$.

1. Consider the case when the deviated strategy $\theta' \in \{\theta_1, \ldots, \theta_{k-1}, \theta_{k+1}, \ldots, \theta_{k_0}\}$. Suppose the player deviates to strategy $k' \neq k$. As a result, the player will utility $w'_{k'}/(m^*_{k'} + 1)$. When $k' \in \mathcal{K}'$, we have that $w'_{k'}/(m^*_{k'} + 1) \leq w'_{k'}/m'_{k'} = z^*$. When $k' \notin \mathcal{K}'$, we have that $w'_{k'}/(m^*_{k'} + 1) = w'_{k'}/(m'_{k'} + 1) < z^*$. Hence, he could get utility at most $z^*$ by deviation.

2. Consider the case when the deviated strategy $\theta' \notin \{\theta_1, \theta_2, \ldots, \theta_{k_0}\}$. Note that $m^*_k \geq m'_k - 1 \geq 2$ by the construction of $m^*_k$. As a result, for any strategy $\theta_{\tilde{k}}$ with $\tilde{k} \in [k_0]$, at least one player chooses it even if player $i$ deviates to $\theta'$. As a result, player $i$ could get utility at most $\sum_{k=k_0+1}^{K} w_k < z^*$.

To conclude, in the strategy profile $\hat{\boldsymbol{\theta}}^*$, every player obtains a utility of at least $z^*$ and can achieve at most $z^*$ by deviating. Therefore, $\hat{\boldsymbol{\theta}}^*$ is a PNE.

(2) We then show that for any PNE $\hat{\boldsymbol{\theta}} = (\hat{\theta}_1, \ldots, \hat{\theta}_N)$, we must have $\forall n \in [N], \hat{\theta}_n \in \{\theta_1, \theta_2, \ldots, \theta_{k_0}\}$. To prove this claim, we have several steps.

*I:* We show that for all $k \in [k_0]$, there exists $i \in [N]$ such that $\hat{\theta}_i = \theta_k$. We prove this by contradiction. Suppose that there exists $k \in [k_0]$ such that for all $i \in [N]$, $\hat{\theta}_i \neq \theta_k$. Since the sum of the utilities of all players is 1, there must exist a player $j$ such that $u_j(\hat{\boldsymbol{\theta}}) \leq 1/N$. Now let $\theta' = \theta_k$. Since all other players do not choose $\theta_k$, player $j$ could become the only player that achieves the minimal loss on data source $k$. As a result, $u_j(\theta', \hat{\boldsymbol{\theta}}_{-j}) = w_k$. Note that

$$
\begin{aligned}
w_k \cdot N \geq w_{k_0} \cdot N &\geq \sum_{k=1}^{k_0} w_{k_0} \cdot \left\lfloor \frac{3w_k'}{w_{k_0}'} \right\rfloor \\
&\geq \sum_{k=1}^{k_0} w_{k_0} \left( \frac{3w_k'}{w_{k_0}'} - 1 \right) \\
&\geq \frac{3w_{k_0}}{w_{k_0}'} - k_0 \cdot w_{k_0} && \text{(Because } \sum_{k=1}^{k_0} w_k' = \sum_{k=1}^{K} w_k = 1) \\
&\geq \frac{3w_{k_0}}{w_{k_0} + \sum_{k'=k_0+1}^{K} w_{k'}} - k_0 \cdot w_{k_0} && \text{(By Eq. (11))} \\
&\geq \frac{3w_{k_0}}{w_{k_0} + w_{k_0}/3} - k_0 \cdot w_{k_0} && \text{(By the assumption } w_{k_0} > 3\sum_{k'=k_0+1}^{K} w_{k'}) \\
&\geq \frac{9}{4} - \sum_{k'=1}^{k_0} w_{k'} && \text{(Because } w_1 > w_2 > \cdots > w_K) \\
&\geq \frac{9}{4} - 1 && \text{(Because } \sum_{k=1}^{K} w_k = 1) \\
&= \frac{5}{4} > 1.
\end{aligned}
$$

Hence, $w_k > 1/N$, implying that player $j$ would achieve a higher utility by deviating to strategy $\theta_k$, leading to a contradiction.

*II:* Suppose there exists at least two players $i, j \in [N]$ such that $\hat{\theta}_i, \hat{\theta}_j \notin \{\theta_1, \ldots, \theta_{k_0}\}$. According to the first step, for any $k \in [k_0]$, there exists at least one player that choose strategy $k$. As a result, the sum of the utilities of players $i$ and $j$ is at most $\sum_{k=k_0+1}^{K} w_k$. Hence, at least one player has utility at most $\sum_{k=k_0+1}^{K} w_k/2$. Without loss of generality, we assume this player is player $i$. Since two players do not choose strategies in $\{\theta_1, \ldots, \theta_{k_0}\}$, there must exist a $k' \in [k_0]$ such that the number of players that choose $\theta_{k'}$ is less than $m_k^*$ (defined in Eq. (17)). Hence, if player $i$ deviates to strategy $\theta_{k'}$, the utility is at least

$$
\frac{w_{k'}}{m_{k'}} \geq \frac{w_{k'}' - \sum_{k=k_0+1}^{K} w_k}{m_{k'}} \geq \frac{w_{k'}'}{m_{k'}} - \frac{\sum_{k=k_0+1}^{K} w_k}{m_{k'}} \geq z^* - \frac{\sum_{k=k_0+1}^{K} w_k}{2} > \frac{\sum_{k=k_0+1}^{K} w_k}{2}.
$$

This leads to a contradiction.

*III:* Suppose there is only one player $i$ that chooses a strategy outside the set $\{\theta_1, \theta_2, \ldots, \theta_{k_0}\}$. The utility of player $i$ is at most $\sum_{k=k_0+1}^{K} w_k < z^*$. In addition, there must exist a $k \in [k_0]$ such that in the PNE, at most $m_k^* - 1$ players choose strategy $\theta_k$. Therefore, if player $i$ deviates to strategy $\theta_k$, he will get utility at least $w_k'/m_k^* \geq w_k'/m_k' \geq z^*$. This leads to a contradiction.

(3) For any PNE $\hat{\boldsymbol{\theta}}$, let $m_k = |\{j \in [N] : \hat{\theta}_j = \theta_k\}|$ be the number of players that choose strategy $\theta_k$ in the PNE. We finally show that $|m_k - m_k'| \leq 1$.

Suppose there exists a $k \in [k_0]$ such that $|m_k - m_k'| \geq 2$. Consider two cases.

1. Consider the case when $m_k - m_k' \geq 2$. Suppose that a player $i$ chooses strategy $\theta_k$. The utility of player $i$ is at most $w_k'/m_k \leq w_k'/(m_k' + 2) < z^*$. Since $\sum_{k=1}^{k_0} m_k = \sum_{k=1}^{k_0} m_k^* = N$ and $m_k \geq m_k' + 2 \geq m_k^* + 2$, there must exist a

$k' \in [k_0], k' \neq k$ such that $m_{k'} < m_{k'}^*$. As a result, if player $i$ deviates to strategy $\theta_{k'}$, he will obtain a utility of at least $w_{k'}'/(m_{k'}+1) \geq w_{k'}'/m_{k'}^* \geq w_{k'}'/m_{k'}' \geq z^* > u_i(\hat{\boldsymbol{\theta}})$, which leads to a contradiction.

2. Consider the case when $m_k' - m_k \geq 2$. Since $\sum_{k=1}^{k_0} m_k = \sum_{k=1}^{k_0} m_k^* = N$ and $m_k^* \geq m_k' - 1 \geq m_k + 1$, there must exist a $k' \in [k_0], k' \neq k$ such that $m_{k'}^* < m_{k'}'$. Let $i$ be any player that chooses strategy $\theta_{k'}$ in the PNE. Then $u_i(\hat{\boldsymbol{\theta}}) = w_{k'}'/m_{k'}' \leq w_{k'}'/(m_{k'}^*+1) \leq z^*$. However, if player $i$ deviates to strategy $\theta_k$, he will obtain a utility of at least $w_k'/(m_k+1) \geq w_k'/(m_k'-1) > z^*$, which leads to a contradiction.

Now the claim follows. $\qquad\square$

### C.6. Proof Sketch of Thm. 5.6

We prove Thm. 5.6 by dividing it into three parts.

**Lemma C.3.** *Under the same assumption as Thm. 5.6, then* $\hat{\boldsymbol{\theta}}^{Homo} = (\hat{\theta}^M, \hat{\theta}^M, \ldots, \hat{\theta}^M)$ *is a PNE if* $t \geq 2\ell_{\max}$.

**Lemma C.4.** *Under the same assumption as Thm. 5.6, then there exists a constant* $\underline{t}$ *such that* $\hat{\boldsymbol{\theta}}^{Homo} = (\hat{\theta}^M, \hat{\theta}^M, \ldots, \hat{\theta}^M)$ *is a PNE if and only if* $t \geq \underline{t}$.

**Lemma C.5.** *Under the same assumption as Thm. 5.6, then there exists a constant* $C > 0$ *such that if* $t \geq \max\{6C/N, 2\ell_{\max}\}$, *then* $\hat{\boldsymbol{\theta}}^{Homo}$ *is the unique PNE.*

Now Thm. 5.6 follows from Lems. C.3 to C.5. The complete proof is available in the full version of the paper at https://arxiv.org/abs/2505.07688.

### C.7. Proof Sketch of Thm. 5.7

We first need the following propositions.

**Proposition C.6.** *Under the same conditions as in Thm. 5.7, there is a constant* $\underline{t} > 0$, *depending only on* $\{\Sigma_k, \theta_k, w_k\}_{k=1}^K$, *such that whenever* $t \leq \underline{t}$, *a strategy profile* $\hat{\boldsymbol{\theta}} = (\hat{\theta}_1, \hat{\theta}_2, \ldots, \hat{\theta}_N) \in \vartheta^N$ *exists with*

$$\left\| \hat{\theta}_n - \hat{\theta}_n^{Prox} \right\|_2 \leq t^2 \quad \text{for all } n \in [N],$$

*and*

$$\left. \frac{\partial u_n(\theta, \hat{\boldsymbol{\theta}}_{-n})}{\partial \theta} \right|_{\theta = \hat{\theta}_n} = 0.$$

We introduce the following constants.

**Definition C.1** ($\ell_D$). *Since* $\theta_i \neq \theta_j$ *for any distinct* $i, j \in [K]$, *we can find a constant* $\ell_D > 0$, *depending only on* $\{\Sigma_k, \theta_k, w_k\}_{k=1}^K$, *such that*

$$\forall \theta \in \vartheta, \quad \left| \left\{ k \in [K] : d_M^2(\theta, \theta_k; \Sigma_k^{-1}) \leq \ell_D \right\} \right| \leq 1.$$

**Definition C.2** ($m_k$). *Let* $m_k = |\{j \in [N] : \hat{\theta}_j^{Prox} = \theta_k\}|$ *be the number of players that choose strategy* $\theta_k$ *in the PNE* $\hat{\boldsymbol{\theta}}^{Prox}$.

Based on Defs. B.1 and C.1, for any $n \in [N]$, we could partition the space $\vartheta$ into four parts.

$$\vartheta_{n,t}^{(1)} = \left\{ \theta \in \vartheta : d_M^2(\theta, \theta_{k_n}; \Sigma_{k_n}^{-1}) \leq t^{3/2} \right\}$$

$$\vartheta_{n,t}^{(2)} = \left\{ \theta \in \vartheta : t^{3/2} < d_M^2(\theta, \theta_{k_n}; \Sigma_{k_n}^{-1}) \leq \ell_D \right\}$$

$$\vartheta_{n,t}^{(3)} = \left\{ \theta \in \vartheta : \forall k \in [K], d_M^2(\theta, \theta_k; \Sigma_k^{-1}) > \ell_D \right\}$$

$$\vartheta_{n,t}^{(4)} = \left\{ \theta \in \vartheta : \exists k \in [K] \backslash \{k_n\}, d_M^2(\theta, \theta_k; \Sigma_k^{-1}) \leq \ell_D \right\}$$

It holds that $\vartheta = \vartheta_{n,t}^{(1)} \cup \vartheta_{n,t}^{(2)} \cup \vartheta_{n,t}^{(3)} \cup \vartheta_{n,t}^{(4)}$. Denote the constant $\underline{t}$ as $\underline{t}_0$ and the strategy profile $\hat{\boldsymbol{\theta}}$ as $\hat{\boldsymbol{\theta}}^{Hete}$ in Prop. C.6.

**Proposition C.7.** *Under the same conditions as in Thm. 5.7, there is a constant* $\underline{t} > 0$ *with* $\underline{t} \leq \underline{t}_0$, *such that when* $t \leq \underline{t}$, *the following holds: for all* $n \in [N]$ *and* $\theta \in \vartheta_{n,t}^{(1)}$, *we have* $u_n(\hat{\boldsymbol{\theta}}^{Hete}) \geq u_n\left(\theta, \hat{\boldsymbol{\theta}}_{-n}^{Hete}\right)$.

**Proposition C.8.** *Under the same conditions as in [Thm. 5.7](), there is a constant $\underline{t} > 0$ with $\underline{t} \leq \underline{t}_0$, such that when $t \leq \underline{t}$, the following holds: for all $n \in [N]$ and $\theta \in \vartheta_{n,t}^{(2)}$, we have $u_n(\hat{\boldsymbol{\theta}}^{Hete}) \geq u_n\left(\theta, \hat{\boldsymbol{\theta}}_{-n}^{Hete}\right)$.*

**Proposition C.9.** *Under the same conditions as in [Thm. 5.7](), there is a constant $\underline{t} > 0$ with $\underline{t} \leq \underline{t}_0$, such that when $t \leq \underline{t}$, the following holds: for all $n \in [N]$ and $\theta \in \vartheta_{n,t}^{(3)}$, we have $u_n(\hat{\boldsymbol{\theta}}^{Hete}) \geq u_n\left(\theta, \hat{\boldsymbol{\theta}}_{-n}^{Hete}\right)$.*

**Proposition C.10.** *Under the same conditions as in [Thm. 5.7](), there is a constant $\underline{t} > 0$ with $\underline{t} \leq \underline{t}_0$, such that when $t \leq \underline{t}$, the following holds: for all $n \in [N]$ and $\theta \in \vartheta_{n,t}^{(4)}$, we have $u_n(\hat{\boldsymbol{\theta}}^{Hete}) \geq u_n\left(\theta, \hat{\boldsymbol{\theta}}_{-n}^{Hete}\right)$.*

Now [Thm. 5.7]() follows directly by combining [Props. C.6]() to [C.10]().

The complete proof is available in the full version of the paper at `https://arxiv.org/abs/2505.07688`.

# D. Important Lemmas

We need the following variants of Farkas's Lemma ([Perng, 2017]()).

**Lemma D.1** (Gordan's Theorem ([Mangasarian](), [1994]())). *For each given matrix $A$, exactly one of the following is true.*

1. *There exists a vector $x$ such that $Ax > 0$.*
2. *There exists a vector $y \geq 0$ and $y \neq 0$ such that $A^\top y = 0$.*

