# OpenReview forum: "Heterogeneous Data Game: Characterizing the Model Competition Across Multiple Data Sources"
_ICML.cc/2025/Conference — ICML 2025 poster_

### Official Review · Reviewer_3G31 · 2025-03-03

**Overall Recommendation:** 3

**Summary:**

This paper investigates the phenomenon of model competition across multiple data sources. The authors propose heterogeneous data game, where each of the model providers decide to deploy a single model, aiming to win the choose from data sources as much as possible. The model is characterized by its parameters and covariance matrix, and the loss incurred when a data source decides to choose a model is measured via Mahalanobis distance. The authors study the cases where the data sources pick models by proximity choice and by probability choice, and prove equivalent condition for pure Nash equilibrium existence when the number of providers $N=2$ in proximity choice case, and give sufficient conditions as well as some basic properties if PNE exists in other cases.

**Claims And Evidence:**

Yes.

**Essential References Not Discussed:**

No.

**Experimental Designs Or Analyses:**

Yes. The experiment focuses on synthetically studying the effect of temperature parameter on existence of Nash equilibrium in the probability choice model of data sources, and is generally intuitive.

**Methods And Evaluation Criteria:**

Yes.

**Other Comments Or Suggestions:**

- I suggest that Assumption 4.1 could be justified more detailedly. The justification in Remark 4.2 gives an intuitive explanation of its satisfiability, while personally I think an elaboration on the case when the covariances are not equal to each other is also needed, since Assumption 4.1 is a key assumption in the rest of the paper.
- In Theorem 5.6, there is an interval between case 1 and case 2. Is the existence of PNE not sured in that interval? The authors would better point out what happened in that interval to make the discussion complete.

**Other Strengths And Weaknesses:**

- Strength: This paper is self-contained. As a theoretical paper, the mathematical modeling of the considered real-world problem is well-defined. Most steps of the abstractions as well as the assumptions in proving theorems are justified. All different cases inside the proposed mathematical model is studied. Generally speaking, this is an interesting paper, and I do not find any serious technical flaw or vulnerability in this research that leading to a straightforward rejection.
- Weakness: I maintain a reserved stance regarding the significance of this paper:
    - The game-theoretical modeling ensures the change of model parameters linearly influence the outcomes, which is a strong simplification of the initial real-world problem.
    - The sufficient conditions are overly intuitive, giving readers a sense of apparent correctness, while the position of necessary conditions, as well as the area between necessity and sufficiency is less explored. And because of this, the practical guide of theoretical results seems not enough, as claimed in abstract.

**Questions For Authors:**

No.

**Relation To Broader Scientific Literature:**

No.

**Theoretical Claims:**

I've only checked the correctness roughly, while I do not discover any technical flaw.

---

> ### Author Rebuttal · Authors · 2025-03-31
>
> Thank you for your reviewing efforts and constructive comments.
>
> > ***Linear Model Assumption***
>
> 1. Despite the prevalence of deep learning, **linear models remain important due to their interpretability—an essential requirement in high-stakes domains** such as healthcare and the judicial system [1]. Additionally, when training data is limited, linear models have been shown to outperform deep learning methods in various settings [2, 3], and thus are still widely used in practice.
> 2. As discussed in the last paragraph of Section 3.3, **our model also aligns with the linear probing paradigm**, where only the final linear layer of a pretrained foundation model is updated. This approach is common when fine-tuning full models is costly or risks overfitting [4]. In this context, our framework can be interpreted as modeling competition over linear heads on fixed representations, offering insights into the emerging market dynamics around foundation model adaptation.
>
> > ***On the Sufficiency, Necessity, and Practical Value of Theoretical Results***
>
> 1. While prior work discusses the existence of homogeneous and heterogeneous PNE, **our main contribution lies in rigorously characterizing *when* they arise under ML-specific conditions**—namely, distribution shifts and high-dimensional strategy spaces. As noted in our response to Reviewer C4o6, most existing models assume low-dimensional spaces with uniform distance metrics, limiting relevance to ML markets.
> 2. **We also further characterize the exact necessary and sufficient condition for homogeneous PNE in HD-Game-Probability** (see our response below on the interval in Theorem 5.6), addressing the gap between sufficiency and necessity.
> 3. **Although the existence of PNE types may seem intuitive, the technical challenge lies in identifying the exact conditions under which they emerge**. The difficulty stems from the high-dimensional continuous strategy space and heterogeneous $\Sigma_k$, which, per Proposition 4.1, yield a non-linear manifold. Analyzing such cases—particularly in HD-Game-Probability (Section 5.3.1)—requires non-standard tools: we first establish a local equilibrium via a fixed-point theorem and then show it is a true PNE using a careful partition of the strategy space (Section C.7).
> 4. From a policy perspective (see Section 1.2), our results offer actionable insights: (1) Theorem 5.4 shows that dominant data sources attract provider focus; our results help quantify how many additional providers or how much incentive is needed to support underrepresented sources. (2) Theorems 5.6 and 5.7 demonstrate that increasing the rationality of data sources can foster heterogeneous equilibria, thus helping to mitigate the risks associated with the dominance of a single model in the market.
>
> > ***Justification of Assumption 4.1***
>
> For any fixed $\theta \in \mathbb{R}^D$, the condition $\bar{\theta}(\mathbf{q}) = \theta$ is equivalent to solving $\sum_{k=1}^K q_k \Sigma_k (\theta_k - \theta) = 0$ under the constraint $\sum_{k=1}^K q_k = 1$. When the covariance matrices $\Sigma_k$ differ, the vectors $\Sigma_k(\theta_k - \theta)$ are typically in general position—especially in high-dimensional settings where $D \gg K$. In such cases, the corresponding overdetermined linear system is injective, and the only solution (except for a measure-zero set with degenerate alignment) is the one uniquely determined by the normalization constraint.
>
> > ***Interval between case 1 and 2 in Theorem 5.6***
>
> We have strengthened our theoretical result to address this point: **In HD-Game-Probability, there exists a threshold $t_0 > 0$ such that the homogeneous PNE exists if and only if $t \ge t_0$.**
>
> The proof uses a constructive argument: assume the homogeneous PNE exists at $t = t'$. For any $t'' > t'$ and deviation $\theta$, the convexity of the Mahalanobis loss implies that the utility of a player deviating to $\theta$ under temperature $t''$ is no greater than the utility of deviating to $c\theta + (1-c)\hat{\theta}^M$ with $c = t'/t''$ under $t'$, which is at most $1/N$. Thus, the deviation is not profitable, and the homogeneous PNE remains valid for all $t'' > t'$, confirming the existence region is an interval $[t_0, \infty)$.
>
> Although the exact value of $t_0$ is hard to compute analytically, our synthetic experiments suggest it is approximately a constant multiple of $2\ell_{\max}$.
>
> ---
>
> [1] Cynthia Rudin. Stop explaining black box machine learning models for high stakes decisions and use interpretable models instead. Nature machine intelligence. 2019.
>
> [2] Shuming Jiao et al. Does deep learning always outperform simple linear regression in optical imaging?. Optics express. 2020.
>
> [3] Muhammad Arbab Arshad et al. The Power Of Simplicity: Why Simple Linear Models Outperform Complex Machine Learning Techniques--Case Of Breast Cancer Diagnosis. 2023.
>
> [4] Ananya Kumar et al. Fine-Tuning can Distort Pretrained Features and Underperform Out-of-Distribution. ICLR. 2022.

---

### Official Review · Reviewer_GPAn · 2025-03-12

**Overall Recommendation:** 3

**Summary:**

The paper explores the HDG problem, aiming to identify the Pure Nash Equilibrium (PNE) in distributing data resources among machine learning (ML) model producers. Each producer provides parameters, influencing resource allocation based on both their own and the ML parameters. The analysis covers conditions for PNE existence in scenarios involving one producer, two producers, and multiple producers.

## update after rebuttal
I keep the score because I think the presentation can be improved.

**Claims And Evidence:**

Yes, all claim has its clear and convincing evidence.

**Essential References Not Discussed:**

Unsure

**Experimental Designs Or Analyses:**

Yes. I have checked the experiment designs and analysis, and I think they are sound and valid.

**Methods And Evaluation Criteria:**

The evaluation criteria are Nash Equilibrium, a well-known notion. For method, the paper focus on proving the existence. They also shown an efficient method to find the lower bound of t in experiment.

It is unclear if Nash equilibrium is good solution concept here. What is the role of pure Nash equilibria in this scenario? How to use it in the real world?

**Other Comments Or Suggestions:**

Maybe dividing each theorem into two or three will make it better. And also the proof is tedious.

**Other Strengths And Weaknesses:**

The paper is technically sound and the result is significant. However, I think the written is not satisfying.

**Questions For Authors:**

In your approach to find heterogeneous PNE in HD-Game-Probability, you start with a PNE for HD-Game-Proximity. However, how you get that PNE? Enumerating all possible profile in (\theta_1, …, \theta_K)^N?

**Relation To Broader Scientific Literature:**

Unsure

**Theoretical Claims:**

Yes. I have checked all proofs and I think they are all correct.

---

> ### Author Rebuttal · Authors · 2025-04-01
>
> Thank you for your reviewing efforts and constructive comments.
>
> > ***It is unclear if Nash equilibrium is good solution concept here. What is the role of pure Nash equilibria in this scenario? How to use it in the real world?***
>
> 1. **Nash equilibrium is a well-established and widely used solution concept in game theory for analyzing strategic interactions among competing agents [1]**. In the context of ML model markets, recent works have also focused on characterizing equilibrium outcomes among model providers [2, 3]. Beyond ML model markets, the use of Nash equilibrium extends to other competitive domains such as recommender systems [4] and targeted advertising [5], where it is employed to model stable market outcomes. Therefore, applying the concept of Nash equilibrium in our setting is both standard and well-supported in the literature.
> 2. **In our model, a pure Nash equilibrium represents a stable state of the market**, where each model provider (player) chooses a strategy (i.e., model parameters) such that no one can unilaterally deviate to improve their utility. This notion of stability is central to real-world markets, where firms adapt until no player can gain from further adjustments, given the choices of others. Thus, analyzing PNE offers insight into the long-term strategic behavior and positioning of providers in a competitive ML ecosystem.
> 3. **Beyond theoretical interest, studying the structure of Nash equilibria can offer practical guidance for market design and policy-making**. As discussed in the last paragraph of Section 1.2, our results provide actionable insights:
>    -  When certain data sources have dominant weights, providers tend to focus on them exclusively (Theorem 5.4). Our analysis quantifies how adding more providers or incentivizing attention to underrepresented sources can shift the equilibrium toward more balanced coverage.
>    - To avoid the risks of model monoculture (i.e., all data sources converging to the same model), Theorems 5.6 and 5.7 show that increasing the rationality of data sources—i.e., improving their ability to select high-performing models—encourages heterogeneous equilibria, leading to more diverse model deployment across the ecosystem.
>
> > ***I think the written is not satisfying. Maybe dividing each theorem into two or three will make it better. And also the proof is tedious.***
>
> 1. We thank the reviewer for the valuable suggestion. **We will revise the paper to improve clarity, organization, and overall readability.** In particular, we will consider breaking down complex theorems—such as Theorems 5.4 and 5.6—into smaller components to make them easier to follow. If there are specific theorems or sections you found especially difficult to read, we would greatly appreciate further feedback, which would help us target our revisions more effectively.
> 2. **Regarding the proofs, we acknowledge that some arguments are lengthy, but this reflects the non-trivial nature of the results.** The main technical challenges arise from the high-dimensional continuous strategy space and heterogeneous covariance matrices $\Sigma_k$, which together form a non-linear manifold of PNE strategies (Proposition 4.1). This complexity is particularly evident in HD-Game-Probability (Section 5.3.1), where we first use a fixed-point theorem to establish a local equilibrium, and then, via a structured partition of the strategy space (Section C.7), show that it satisfies full PNE conditions. We will revise the proofs to improve clarity and provide more intuitive explanations in the future version of the paper.
>
> > ***How you get that PNE? Enumerating all possible profile in $(\theta_1, ..., \theta_K)^N$?***
>
> **The initial PNE for HD-Game-Proximity is derived using Theorem 5.4 and Corollary 5.5.** These results show that, at equilibrium, the number of model providers focusing on each data source is approximately proportional to that source’s weight. The specific allocation is computed using Equation (11), which determines how many providers should select each $\theta_k$.
>
> **This method provides an efficient way to construct a candidate PNE and does not rely on exhaustive enumeration over all possible strategy profiles in $(\theta_1, ..., \theta_K)^N$.** As a result, it is computationally efficient and scalable in our experimental setup.
>
> ---
>
> [1] Noam Nisan et al. Algorithmic game theory. Cambridge Univ. Press. 2007.
>
> [4] Omer Ben-Porat et al. Regression equilibrium. ACM Conference on Economics and Computation. 2019.
>
> [3] Meena Jagadeesan et al. Improved bayes risk can yield reduced social welfare under competition. NeurIPS. 2023.
>
> [4] Meena Jagadeesan et al. Supply-side equilibria in recommender systems. NeurIPS 2023.
>
> [5] Ganesh Iyer et al. Competitive model selection in algorithmic targeting. Marketing Science. 2024.

---

### Official Review · Reviewer_1w5x · 2025-03-13

**Overall Recommendation:** 4

**Summary:**

This work analyze competition through Nash equilibria between multiple ML model providers across heterogeneous data sources. The game is characterized under two different data source choice models and provides conditions for each type of equilibrium. Synthetic experiments are conducted.

**Claims And Evidence:**

Characterization of conditions for PNE existence is supported with theoretical analysis.
Stronger evidence is the practical generalizability beyond linear models.

**Essential References Not Discussed:**

N/A

**Experimental Designs Or Analyses:**

Only experiments with synthetic data are conducted, which validate theoretical analysis.

**Methods And Evaluation Criteria:**

Theoretical findings are evaluated with synthetic experiments for various parameters such as model choice temperature and number of model providers. Experiments are rather basic and additional real-world examples would have strengthened the evaluation.

**Other Comments Or Suggestions:**

* In line 128 "There are N model providers (players) that need to compete the models in these K data sources."
* Explain how a data source queries each model provider for $\ell_{n,k}$. Is this validation loss of each data source? Mention why assuming this is available is reasonable for a data market setting.

**Other Strengths And Weaknesses:**

A limitation of this analysis is focused on linear models and IID distributions. Limited empirical validation with real data.

**Questions For Authors:**

* How do the model providers optimize $\hat{\theta}$ in practice?
* How do the model providers learn their loss on each data source? This is not discussed.
* Does this assume stationarity of data source distribution? How would distribution shifts affect equilibria. How robust are your equilibrium results to non-stationarity in data distributions?

**Relation To Broader Scientific Literature:**

The related section seems comprehensive and their work is well-situated in the literature. The main novelty seems to be extending the study of equilibrium from homogeneous data source to heterogeneous data source.

**Theoretical Claims:**

Statements in the main text look right. Proofs in the appendix were not checked.

---

> ### Author Rebuttal · Authors · 2025-03-31
>
> Thank you for your reviewing efforts and constructive comments.
>
> > ***Focus on Linear Models, IID Assumptions, and Lack of Empirical Validation***
>
> 1. **Linear Models**: Despite the rise of deep learning, linear models remain widely used for their interpretability, particularly in high-stakes domains [1], and can outperform deep models when data is limited [2]. Our framework also aligns with the linear probing setup, where only the final layer of a foundation model is updated—common when full fine-tuning is costly or prone to overfitting [3]. This makes our model relevant for analyzing competition in foundation model markets.
> 2. **IID Assumption**: Our setting is inherently non-IID, as data sources follow different distributions, reflected by distinct covariance matrices $\Sigma_k$. This heterogeneity is central to our analysis. Extending to more complex or dynamic distributions is a promising direction for future work.
> 3. **Empirical Validation**: Our synthetic experiments are designed to rigorously verify our theoretical results and explore cases beyond analytical coverage. Synthetic data allows us to exhaustively check whether a strategy profile constitutes a PNE, which is difficult to do reliably in real-world high-dimensional settings. While experimental validation in real applications is indeed valuable, we emphasize that our theoretical results apply to such settings and remain relevant even when empirical verification is challenging.
>
> > ***In line 128 "There are N model providers (players) that need to compete the models in these K data sources."***
>
> Thank you for pointing this out. If the concern is about clarity, we are happy to revise the sentence to: "There are $N$ model providers (players), each deploying a single model to compete across $K$ data sources." Please let us know if a different clarification was intended.
>
> > ***Explain how a data source queries each model provider for $\ell_{n,k}$. Is this validation loss of each data source?***
>
> Yes, $\ell_{n,k}$ can be interpreted as the validation loss of model provider $n$ on data source $k$. The availability of such loss information is a standard modeling assumption in prior work on ML model competition [4].
>
> In practice, this value can be obtained through interactions between model providers and data sources. A common scenario is: the data source shares a private validation set with model providers, who return predictions. The data source then evaluates the loss and may report it back to the provider.
>
> > ***How do the model providers optimize $\hat{\theta}$ in practice?***
>
> 1. Our focus is on analyzing the properties of equilibrium among model providers, representing the stable market states. Nash equilibrium is a standard and widely used solution concept in both ML model markets [4] and broader competitive settings [5].
> 2. The process by which providers optimize $\hat{\theta}$ is an interesting direction for future work. While not the focus of our paper, a common approach—also used in prior work on ML model markets without heterogeneous data [6]—is **best-response dynamics**, where each provider updates $\hat{\theta}_n$ to maximize their utility in Equation (3), given others' strategies. A formal analysis of convergence in our setting remains open.
>
> > ***How do the model providers learn their loss on each data source?***
>
> As noted above, model providers can obtain loss information through interactions with data sources. For example, a data source may provide a validation set, and after receiving predictions from the provider, compute and share the resulting loss. This feedback allows providers to estimate $\ell_{n,k}$, which in turn guides how they weight different data sources during model training.
>
> > ***Stationarity Assumption and Robustness to Distribution Shifts***
>
> Our current analysis assumes fixed (stationary) data distributions. However, the equilibrium is robust to moderate distributional shifts, as key parameters (e.g., $\ell_{\max}$, $\hat{\theta}^M$) vary smoothly with the distribution. Thus, small changes lead to small adjustments in PNE. Thus, our theoretical insights continue to hold under mild distributional shifts. Studying dynamic or non-stationary settings remains an important direction for future work.
>
> ---
>
> [1] Cynthia Rudin. Stop explaining black box machine learning models for high stakes decisions and use interpretable models instead. Nature machine intelligence. 2019.
>
> [2] Muhammad Arbab Arshad et al. The Power Of Simplicity: Why Simple Linear Models Outperform Complex Machine Learning Techniques--Case Of Breast Cancer Diagnosis. 2023.
>
> [3] Ananya Kumar et al. Fine-Tuning can Distort Pretrained Features and Underperform Out-of-Distribution. ICLR. 2022.
>
> [4] Omer Ben-Porat et al. Regression equilibrium. ACM Conference on Economics and Computation. 2019.
>
> [5] Noam Nisan et al. Algorithmic game theory. Cambridge Univ. Press. 2007.
>
> [6] Omer Ben-Porat et al. Best response regression. NeurIPS. 2017.

---

### Official Review · Reviewer_C4o6 · 2025-03-21

**Overall Recommendation:** 2

**Summary:**

This work considers a game between model providers who choose which data sources to include in their models training. The approach is one like facility location games: the data sources are also the customers, and so e.g. a model provider could prioritize one data source to ensure that they win business from that customer. Or, in the presence of no competition, the model provider could equally weight all data sources, or weight them according to their sizes within the market. The example up front given is hospitals: they are both providing the data for training the algorithms, and they are the consumers of medical models that are trained on data from their and other hospitals.

The paper considers pure Nash equilibria, and whether or not equilibria: a) do not exist, b) exist where all model providers converge to same outcome, or c) where model providers specialize on different sources or mixtures of sources. These are considered under two regimes: in which the facilities / data providers choose exactly the best model provider, or noisily choose providers.

With two competing firms, the firms converge to the dominant data source if there is one; otherwise, there is no pure Nash equilibrium. However, this is called Heterogenous for some strange reason. I do not understand that.
With more than two competing firms, the paper gives conditions on PNE existence that translate to usually finding heterogeneous strategies when data source provider is chosen optimally, and can be heterogenous or homogeneous under noisily chosen parameters. Examples and characterizations of the spaces that lead to either are included.

**Claims And Evidence:**

I did not review the proofs in detail. The claims do not seem to be problematic.

**Essential References Not Discussed:**

Not to my knowledge.

**Experimental Designs Or Analyses:**

The paper simulates these games experimentally, but the results are just aggregated by showing 10 sample runs, not any statistics aggregated over the outputs or treatment of whether these are representative.

**Methods And Evaluation Criteria:**

It is helpful that the methods include both theoretical proofs and examples.

**Other Comments Or Suggestions:**

Nit to pick: In 3.2, notationally, you should be defining g, not g^{prop,prox}. You could though say g = g^prop = ....

**Other Strengths And Weaknesses:**

The topic is interesting, and facility location makes for a good literature to draw on. I don't know that the results and model formulation are of the utmost interest. It seems clear that sometimes firms would consolidate, and sometimes differentiate. For a full treatment, this should be compared to other results in the facility location literature and general economics competition literature - what is different about the existence, homogeneous vs heterogeneous results in this setting vs other settings?  The paper speaks to a desire to have policy impacts, but the results are not discussed at the level that would result in policy discussions. What are the policy implications here? How should a policy maker evaluate this in the example scenario of training on hospital data?

The paper is poorly written, which makes it hard to follow. Here are some examples.
- Proposition 4.1 is described as characterizing the "strategy set" for each player, but it is really talking about best response strategies.
- In the definition of The Heterogeneous Data Game, the type parameter \theta on the left hand side of the equation does not show up on the right side, it is implicit in l_{n,k}. This is confusing. Adjust your notation so that you have enough space to include the type explicitly.
- g_n(l_{1,k}, ..., l_{n,k}) is the weight for facility k on data provider n, which must satisfy \sum_{n\ in N} g_n(l_{.., k})=1. That seems strange, for it to be a choice of k and not n. Perhaps put a k superscript on g. Then you could say: g_n^k(  l(theta hat) ). That notation makes much more sense.

**Questions For Authors:**

How do your heterogeneous and homogeneous equilibrium results compare to the findings in the facility location? If a policy maker takes their intuition from general facility location games, what do they lose out on vs understanding the model you present?

**Relation To Broader Scientific Literature:**

This paper proposes a model based on facility location, which has an extensive literature. That relationship is an interesting one to draw on.  Those connections should be further discussed - what are the closest results? Is the intuition similar or different when moving from euclidean space into ML model competition?

**Theoretical Claims:**

I did not verify the correctness of the proofs in the supplemental appendix.

---

> ### Author Rebuttal · Authors · 2025-03-31
>
> Thank you for your reviewing efforts and constructive comments.
>
> > ***Comparison with Competitive Location Models***
>
> **1. Differences in Setting**
> Our model captures two key features of ML markets often missing in prior work: (1) **Source-specific distance metrics** from distribution shifts, and (2) **High-dimensional strategy spaces** due to many sources and model parameters.
>
> In contrast, most competitive location models focus on **low-dimensional spaces** or **networks** with **uniform distance metrics**, driven by two factors: (1) applications in urban planning naturally fit 1D, 2D, or network settings; and (2) many models involve additional variables like price or quantity, limiting tractability and leading to smaller-scale formulations.
>
> While a few papers explore high-dimensional competition, they do so in the context of quantity competition [1] or pricing [2], rather than spatial or parameter competition.
>
> **2. Technical Distinctions and Contributions**
> While prior models also observe homogeneous and heterogeneous equilibria, our key contribution lies in characterizing *when* they arise under ML-specific conditions.
>
> - **Heterogeneous distance metrics**: When $\Sigma_k$ differ across sources, Proposition 4.1 shows that the PNE strategy set becomes a non-linear manifold in $\mathbb{R}^D$, making equilibrium analysis substantially more complex.
> - **High-dimensionality**: To our knowledge, no prior work studies PNE in high-dimensional continuous spaces, even under Euclidean distance. As prior results rely heavily on geometric properties [3, 4], existing theories do not extend to our setting.
>
> We acknowledge connections to low-dimensional results. For example, [3] analyzes a 2D probabilistic model with conditions for homogeneous PNE, while [5] studies location–quality competition, showing that proximity choice leads to homogeneous PNE, and probabilistic choice allows both types of PNE. However, these insights do not extend to **high-dimensional settings with heterogeneous metrics**. Classical economic models (e.g., Bertrand, Cournot) focus on price or quantity competition under simpler assumptions and do not capture the geometric complexity introduced by heterogeneous loss functions in our setting.
>
> Technically, analyzing PNE requires new tools. To prove Theorem 5.7, we apply a fixed-point theorem to find a local equilibrium and show, via a partition of the strategy space (Section C.7), that it satisfies PNE conditions. The analysis is non-trivial due to the geometric complexity involved.
>
> > ***Policy Implications***
>
> **Our model captures key aspects of ML markets overlooked in prior work, as previously noted.** As discussed in Section 1.2, our theoretical results offer the following novel policy insights:
>
> 1. When certain data sources (e.g., hospitals) have dominant weights, providers tend to focus on them (Theorem 5.4). Policymakers can counter this by adding more providers or incentivizing attention to smaller sources; our results help quantify both.
> 2. To promote model diversity and avoid convergence to a single dominant model (e.g., in medical ML), Theorems 5.6 and 5.7 show that increasing the rationality of data sources—i.e., improving their ability to choose better models—encourages heterogeneous PNE.
>
> > ***Writing Issues***
>
> We thank the reviewer for the helpful suggestions and will revise the paper accordingly. In addition, in HD-Game-Proximity with $N=2$, we label the PNE as "heterogeneous" since the strategy is $(\theta_1,\theta_1)$, not the homogeneous PNE given in the paper, consistent with the heterogeneous structure observed for $N>2$.
>
> > ***Justification of Experiments***
>
> Our synthetic experiments verify theoretical results and explore scenarios beyond theories. Ten representative simulations align well with theory. We further ran 100 random instances and report the average and standard deviation of temperature thresholds, confirming, for example, that the minimal temperature for a homogeneous PNE is typically well below $2\ell_{\max}$.
>
> |N|5|10|15|20|25|30|
> |:--|:--|:--|:--|:--|:--|:--|
> |Homogeneous PNE: Minimal $t/(2\ell_{\max})$|0.11 (0.07)|0.15 (0.08)|0.17 (0.08)|0.17 (0.08)|0.18 (0.09)|0.18 (0.09)|
> |Hetegeneous PNE: Maximal $t/(2\ell_{\max})$|0.09 (0.07)|0.12 (0.07)|0.12 (0.07)|0.11 (0.07)|0.11 (0.07)|0.11 (0.07)|
>
> ---
>
> [1] Simon P. Anderson et al. Spatial Competition a la Cournot: Price Discrimination by Quantity‐Setting Oligopolists. Journal of Regional Science. 1990.
>
> [2] Helmut Bester. Noncooperative bargaining and spatial competition. Econometrica. 1989.
>
> [3] Dodge Cahan et al. Spatial competition on 2-dimensional markets and networks when consumers don't always go to the closest firm. International Journal of Game Theory. 2021.
>
> [4] Zvi Drezner et al. Competitive location models: A review. European Journal of Operational Research. 2024.
>
> [5] M. Elena Sáiz et al. On Nash equilibria of a competitive location-design problem. European Journal of Operational Research. 2011.

---

### Decision · Program_Chairs · 2025-05-01

**Decision:**

Accept (poster)

**Comment:**

This paper considers a data market where multiple ML providers compete for heterogeneous data sources, heterogeneity being the key novelty. Under various conditions, the paper establishes what equilibria the ML providers will converge to, if any (e.g., all the same model, or each a specialized model). Reviewers had a favorable view of the paper. The main feedback is that the writing and exposition can be improved. This includes clearly delineating the setting of the paper from others, improving the mathematical language, and clearly articulating the motivation/description of experiments (possibly adding non-synthetic examples). Some of the limitations of the setting should also be better discussed (e.g., linear models). We urge the authors to closely follow all the suggestions of the reviewers and improve the clarity of the paper.